# CKAMP44 modulates integration of visual inputs in the lateral geniculate nucleus

Xufeng Chen[1,2], Muhammad Aslam[1,2], Tim Gollisch [3], Kevin Allen[4] & Jakob von Engelhardt[1,2]

Relay neurons in the dorsal lateral geniculate nucleus (dLGN) receive excitatory inputs from retinal ganglion cells (RGCs). Retinogeniculate synapses are characterized by a prominent short-term depression of AMPA receptor (AMPAR)-mediated currents, but the underlying mechanisms and its function for visual integration are not known. Here we identify CKAMP44 as a crucial auxiliary subunit of AMPARs in dLGN relay neurons, where it increases AMPAR-mediated current amplitudes and modulates gating of AMPARs. Importantly, CKAMP44 is responsible for the distinctive short-term depression in retinogeniculate synapses by reducing the rate of recovery from desensitization of AMPARs. Genetic deletion of CKAMP44 strongly reduces synaptic short-term depression, which leads to increased spike probability of relay neurons when activated with high-frequency inputs from retinogeniculate synapses. Finally, in vivo recordings reveal augmented ON- and OFF-responses of dLGN neurons in $CKAMP44$ knockout ($CKAMP44^{-/-}$) mice, demonstrating the importance of CKAMP44 for modulating synaptic short-term depression and visual input integration.

[1] Institute of Pathophysiology, University Medical Center of the Johannes Gutenberg University Mainz, Mainz 55128, Germany. [2] Synaptic Signalling and Neurodegeneration, German Center for Neurodegenerative Diseases (DZNE), Bonn 53127, Germany. [3] Department of Ophthalmology, University Medical Center Göttingen, Göttingen 37037, Germany. [4] Department of Clinical Neurobiology, Medical Faculty of Heidelberg University and German Cancer Research Center (DKFZ), 69126 Heidelberg, Germany. Correspondence and requests for materials should be addressed to J.v.E. (email: engelhardt@uni-mainz.de)

Relay neurons of the dLGN transform input from the retina so that spikes with high visual information content are preferentially transmitted to the cortex[1]. Spike timing of RGCs is very important for spike transmission with a sharp increase in the transmission rate for spike frequencies of more than 30 Hz[2] suggesting that synaptic short-term plasticity plays a crucial role for input integration[3–5]. Retinogeniculate synapses display a pronounced short-term depression that arises partly from a high release probability[6]. Interestingly, desensitization of AMPARs contributes to the short-term depression in this synapse[7]. That is unusual as in most other synapses, release probability is too low and recovery from desensitization of AMPARs too fast for a significant contribution of AMPAR desensitization to short-term plasticity[8].

A combination of several functional and morphological properties explains why desensitization of AMPARs plays a role in short-term depression at retinogeniculate synapses. Firstly, RGC axons contact dLGN neurons with large terminals that comprise multiple neighboring release sites[9]. This geometry precludes fast diffusion of glutamate out of the synaptic cleft and allows spillover of glutamate to non-active neighboring synapses such that AMPARs of active and non-active synaptic sites of a given terminal are strongly desensitized[10]. Secondly, AMPARs of retinogeniculate synapses display pronounced desensitization and slow recovery from desensitization, causing long-lasting depression of AMPAR-mediated currents[10].

Gating kinetics of AMPARs in dLGN neurons, in particular the very slow recovery from desensitization, are very different from that of heterologously expressed tetrameric receptors[11]. This suggests that auxiliary subunits influence AMPAR function in dLGN neurons. Auxiliary subunits such as TARPs, cornichons, CKAMPs, and GSG1L are expressed throughout the brain and exert a strong influence on AMPAR trafficking and gating[12–18]. The prototypical auxiliary subunit stargazin influences AMPAR-mediated currents in dLGN relay neurons[19]. However, it renders recovery from desensitization faster[12], suggesting that another auxiliary subunit is responsible for the unusual AMPAR gating kinetics in dLGN relay neurons. Three of the known auxiliary subunits decrease the rate of recovery from desensitization, namely CKAMP39, CKAMP44, and GSG1L[15,18,20]. In situ hybridization experiments, however, show that GSG1L and CKAMP39 are not highly expressed in in the thalamic region[18,21], suggesting that CKAMP44 may influence AMPAR function in this brain region.

Using in vitro and in vivo electrophysiology, we here show that CKAMP44 indeed influences the number and gating of synaptic and extrasynaptic AMPARs in dLGN neurons. Importantly, the genetic deletion of CKAMP44 increases the rate of recovery from desensitization and reduces short-term depression in retinogeniculate synapses. By its influence on short-term plasticity, CKAMP44 plays a role for input processing as evidenced by increased excitatory postsynaptic potential (EPSP) amplitudes and spike probability upon repetitive stimulation of the optic tract in CKAMP44$^{-/-}$ mice. Findings from in vitro slice experiments were corroborated by in vivo unit activity recordings showing that dLGN ON- and OFF-responses are increased in CKAMP44$^{-/-}$ mice.

## Results

**CKAMP44 regulates surface expression and gating of AMPARs**. In situ hybridization experiments using oligoprobes had suggested that CKAMP44 may be expressed in the dLGN although the signal intensity was weaker than that in the dentate gyrus (Fig. 1a and ref. [18]). However, the lower neuron density in the dLGN as compared to that in the dentate gyrus may lead to an underestimation of the signal intensity per cell. Indeed, in situ hybridization using riboprobes revealed a CKAMP44 mRNA signal intensity that was similar if not higher in relay neurons of the dLGN when compared to the signal intensity in dentate gyrus granule cells (Fig. 1b). This suggests that CKAMP44 is expressed in the dLGN and may thus influence AMPAR function there.

To investigate the influence of CKAMP44 on synaptic AMPARs, we recorded AMPAR-mediated currents of retinogeniculate and corticogeniculate synapses by stimulating the optic tract or corticogeniculate fibers, respectively (Fig. 1c, d). The amplitude of AMPAR-mediated currents was normalized to that of NMDAR-mediated currents recorded at a holding potential of +40 mV. The AMPA/NMDA ratio was significantly reduced in both synapses of CKAMP44$^{-/-}$ mice (Fig. 1c, d and Supplementary Table 1), indicating that deletion of CKAMP44 reduces the number of AMPARs in retinogeniculate and corticogeniculate synapses. To provide further evidence that CKAMP44 is present in both synapse types of wildtype mice, we analyzed the decay time constant ($\tau_{decay}$) of evoked AMPAR-mediated currents in the presence of cyclothiazide (CTZ), which blocks AMPAR desensitization and decreases the rate of deactivation. As shown previously, CTZ increases the $\tau_{decay}$ considerably more for AMPARs that interact with CKAMP44 than compared to CKAMP44-less AMPARs[18]. In the absence of CTZ, the $\tau_{decay}$ of AMPAR-mediated currents in retinogeniculate and corticogeniculate synapses did not differ between wildtype and CKAMP44$^{-/-}$ mice (Supplementary Fig. 1a, c and Supplementary Table 2). However, CTZ indeed increased the $\tau_{decay}$ of AMPAR-mediated currents considerably more in both synapse types of wildtype than in that of CKAMP44$^{-/-}$ mice (Supplementary Fig. 1b, d and Supplementary Table 2). This suggests that CKAMP44 binds to AMPARs in retinogeniculate and corticogeniculate synapses.

To investigate the influence of CKAMP44 on AMPAR gating kinetics, we evoked extrasynaptic AMPAR-mediated currents by ultra-fast application of glutamate onto nucleated patches of relay neurons (Fig. 1e). The current amplitude was reduced in CKAMP44$^{-/-}$ mice to a similar extent as the AMPA/NMDA ratio (Fig. 1f), indicating that deletion of CKAMP44 reduced the number of extrasynaptic and synaptic AMPARs. Deletion of CKAMP44 did not alter the time constants of deactivation ($\tau_{deact}$) and desensitization ($\tau_{desen}$) or the rise time of AMPAR-mediated currents, but increased the steady-state current amplitude (as a percentage of the peak amplitude) (Fig. 1f, g and Supplementary Table 3). Importantly, the recovery from desensitization analyzed with two 1-ms glutamate pulses with different inter-stimulus intervals was considerably faster in CKAMP44$^{-/-}$ mice than in wildtype mice as evidenced by a significant reduction in the weighted time constant of recovery ($\tau_{recovery}$) (Fig. 1h and Supplementary Table 3).

**CKAMP44 influences retinogeniculate short-term plasticity**. AMPAR desensitization has previously been shown to be partly responsible for the pronounced short-term depression in this synapse. Thus, the strong modulation of the recovery from desensitization exerted by CKAMP44 suggested that the auxiliary subunit might influence short-term plasticity in retinogeniculate synapses. Indeed, the paired-pulse ratio (PPR) of AMPAR-mediated currents was significantly increased in CKAMP44$^{-/-}$ mice when stimulating the optic tract twice with different inter-stimulus intervals (30 ms, 100 ms, 300 ms, 1000 ms, 3000 ms) (Fig. 2a and Supplementary Table 4). In corticogeniculate synapses, deletion of CKAMP44 did not significantly affect the PPR (Fig. 2b and Supplementary Table 4), consistent with a small contribution of AMPAR desensitization to short-term plasticity in this low-release probability synapse[22].

Note that the voltage-clamp experiments were performed with a cesium (Cs⁺)-based solution in the patch pipette, and it has been reported that cesium (Cs⁺) may increase presynaptic vesicle release probability[23]. Thus, it might be argued that the use of cesium, which is blown onto the slice during patching procedure, might augment the influence of CKAMP44 on PPR. However, when using a potassium (K⁺)-based intracellular solution rather than a Cs⁺-based solution, PPRs were still significantly higher in

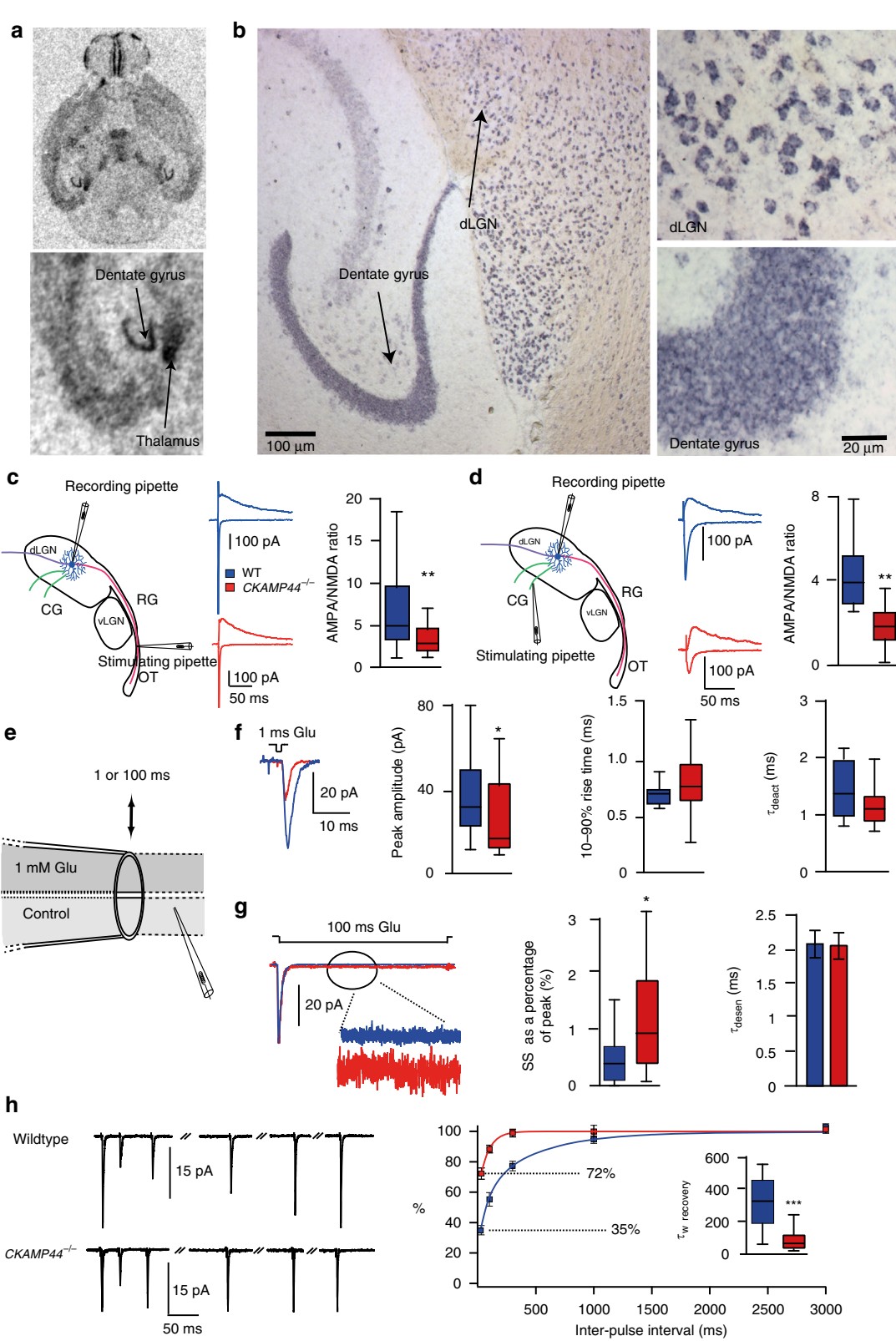

retinogeniculate synapses of $CKAMP44^{-/-}$ mice than in those of wildtype mic (Fig. 2a, and Supplementary Fig. 2, Supplementary Tables 4 and 5).

Transmission of information from the retina to the dLGN is most efficient when RGCs fire with high frequency of more than 30 Hz[2]. Deletion of CKAMP44 increased PPR especially for short inter-stimulus intervals suggesting that the influence of the AMPAR auxiliary subunit on short-term depression is relevant for input integration. However, short-term plasticity-mediated changes in synaptic strength might also be relevant for signal transmission when RGCs fire with lower frequencies[2]. We tested how synaptic strength changes when retinogeniculate fibers are activated several times at lower frequencies. To this end, we stimulated the optic tract with a train of stimuli (40 times) with different frequencies (1, 3.3, and 10 Hz). Figure 2c, left panel, shows examples of AMPAR-mediated currents with 10 Hz stimulation. The current amplitudes were normalized to the first current amplitude and fitted with an exponential function to obtain an estimate of the steady-state EPSC amplitude (Fig. 2c middle panel). EPSC amplitudes reduced mainly in the beginning of the stimulus train. This reduction was significantly smaller in $CKAMP44^{-/-}$ than in wildtype mice when stimulating with 3.3 and 10 Hz (Fig. 2c right panel and Supplementary Table 6), suggesting that CKAMP44 reduces the synapse strength when RGCs display background-firing frequencies larger than 1 Hz.

As shown above, deletion of CKAMP44 reduces synaptic AMPAR-mediated current amplitude by 43% (most likely due to a reduced number of AMPARs; Fig. 1c, d and Supplementary Table 1). However, this reduction was estimated from the reduced AMPA/NMDA ratio, which was obtained by stimulating axons at very low frequency. The difference in synapse strength between $CKAMP44^{-/-}$ and wildtype mice will reduce with increasing RGC background-firing frequencies. Thus, to get a more realistic estimate about changes in synaptic strength during different RGC background-firing frequencies, we took the reduction in synapse strength (i.e., reduction in AMPAR number) in $CKAMP44^{-/-}$ mice into account and shifted the steady-state EPSC amplitude curve accordingly (Supplementary Fig. 3). With this correction, steady-state EPSC amplitudes are similar in both genotypes when stimulating with 10 Hz. This suggests that the strength of retinogeniculate synapses of both genotypes is comparable despite the reduced number of AMPARs in synapses of $CKAMP44^{-/-}$ mice when RGCs fire regularly at ~10 Hz.

**CKAMP44 decreases the firing probability of relay cells**. Voltage-clamp experiments showed that CKAMP44 increases short-term depression in retinogeniculate synapses by slowing recovery from desensitization of AMPARs. To further investigate

if deletion of CKAMP44 affects integration of excitatory inputs in this synapse, we performed experiments in current-clamp mode and stimulated the optic tract ten times at 50 Hz. Relative EPSP amplitudes showed strong depression in relay neurons of wildtype mice, but remained by and large constant in $CKAMP44^{-/-}$ mice (Fig. 3a and Supplementary Table 7). The example traces displayed in Fig. 3a show that EPSP summation results in absolute EPSP response amplitudes that are increasingly different between genotypes with each consecutive stimulus. The changes in relative EPSP amplitudes in wildtype and $CKAMP44^{-/-}$ mice was comparable to the changes in EPSC amplitudes when analyzed with the same stimulation protocol and intracellular solution (i.e., $K^+$-based) (Fig. 3a, Supplementary Fig. 4, Supplementary Tables 7 and 8).

The difference of EPSP amplitudes was not due to changes in EPSP kinetics or resting membrane potential (Supplementary Fig. 5, Supplementary Fig. 6, Supplementary Tables 9 and 10). Moreover, we stimulated such that the amplitude of the first EPSP was similar (Supplementary Fig. 5a and Supplementary Table 7) to exclude that differences in the activation of voltage gated channels influences the result. The difference in EPSP amplitudes could also not be explained by a change in the contribution of NMDAR- or $GABA_AR$-mediated currents, as a similar difference was observed in the presence of the antagonists APV and SR 95531 hydrobromide (gabazine) (Supplementary Fig. 5b, c and Supplementary Table 7).

To investigate if CKAMP44 influences not only EPSP amplitudes, but also firing of relay neurons, we increased the stimulation strength of the 50 Hz stimulus such that the 2nd to 10th stimulus, but not the 1st, often evoked an action potential. Deletion of CKAMP44 then increased spike probability for stimulus 4–10 and the average firing rate when quantified for the 200 ms of the 50 Hz stimulus train (Fig. 3b, Supplementary Tables 11 and 12). This increase was not due to changes in passive and active membrane properties of relay neurons in $CKAMP44^{-/-}$ mice (Supplementary Fig. 6 and Supplementary Table 10). Importantly, spike probability and firing rate were increased in relay neurons of $CKAMP44^{-/-}$ mice despite a 48% smaller first EPSP amplitude (Supplementary Tables 11–13). We adjusted the stimulation strength in this experiment such that the amplitude of the first EPSP was smaller in $CKAMP44^{-/-}$ than in wildtype mice to take the reduced synapse strength in $CKAMP44^{-/-}$ mice into account.

We then analyzed if there is also a difference in spike probability and average firing rate when the optic tract is stimulated with 40 pulses with different frequencies (1, 3.3, and 10 Hz) before the high frequency stimulus train (10 stimuli at 50 Hz). The idea of this experiment was to simulate background

**Fig. 1** CKAMP44 modulates gating and amplitude of AMPAR-mediated currents in dLGN neurons. **a, b** CKAMP44 in situ hybridization analysis with an oligoprobe (**a**) and riboprobe (**b**) on horizontal brain sections of P14 and adult mice respectively **a**: $n = 3$, **b**: $n = 6$). The higher magnification in **b** shows the comparable CKAMP44 mRNA signal intensity in dLGN neurons and dentate gyrus granule cells. **c, d** The AMPA/NMDA ratio is reduced in retinogeniculate (RG;**c**, $n = 19$ for relay neurons of wildtype and 19 for relay neurons of $CKAMP44^{-/-}$ mice) and corticogeniculate (CG; **d** $n = 15$ for relay neurons of wildtype and 18 for relay neurons of $CKAMP44^{-/-}$ mice) synapses of $CKAMP44^{-/-}$ mice (Box-and-whisker Tukey plots with median, IQR and 1.5 times the IQR, Mann–Whitney test). A schematic representation of RG and CG synapse activation with localization of stimulation pipettes is shown on the left of **c**, **d**. **e** Schematic representation of the nucleated patch experiments. **f** The amplitude of extrasynaptic AMPAR-mediated currents is reduced in relay neurons of $CKAMP44^{-/-}$ mice. There is no change in rise time and $\tau_{deact}$ ($n = 18$ for relay neurons of wildtype and 21 for relay neurons of $CKAMP44^{-/-}$ mice, box-and-whisker Tukey plots, Mann–Whitney test). **g** The steady-state current amplitude is increased in dLGN neurons of $CKAMP44^{-/-}$ mice (Box-and-whisker Tukey plots, Mann–Whitney test). The $\tau_{des}$ is not significantly changed ($n = 20$ for relay neurons of wildtype and 17 for relay neurons of $CKAMP44^{-/-}$ mice, mean ± SEM, t-test). **h** The $\tau_{recovery}$ of AMPAR-mediated currents is reduced in relay neurons of $CKAMP44^{-/-}$ mice. Example traces of pairs of currents that were evoked with two glutamate 1 ms pulses with different inter-pulse intervals (30, 100, 300, 1000, and 3000 ms) are shown on the left. The quantification shows the amplitude of the second current as a percentage of the first current ($n = 20$ for relay neurons of wildtype and 17 for relay neurons of $CKAMP44^{-/-}$ mice, box-and-whisker Tukey plots, Mann–Whitney test). The $\tau_{recovery}$ was estimated from a double-exponential fit of the data. *$p < 0.05$, **$p < 0.01$, ***$p < 0.001$

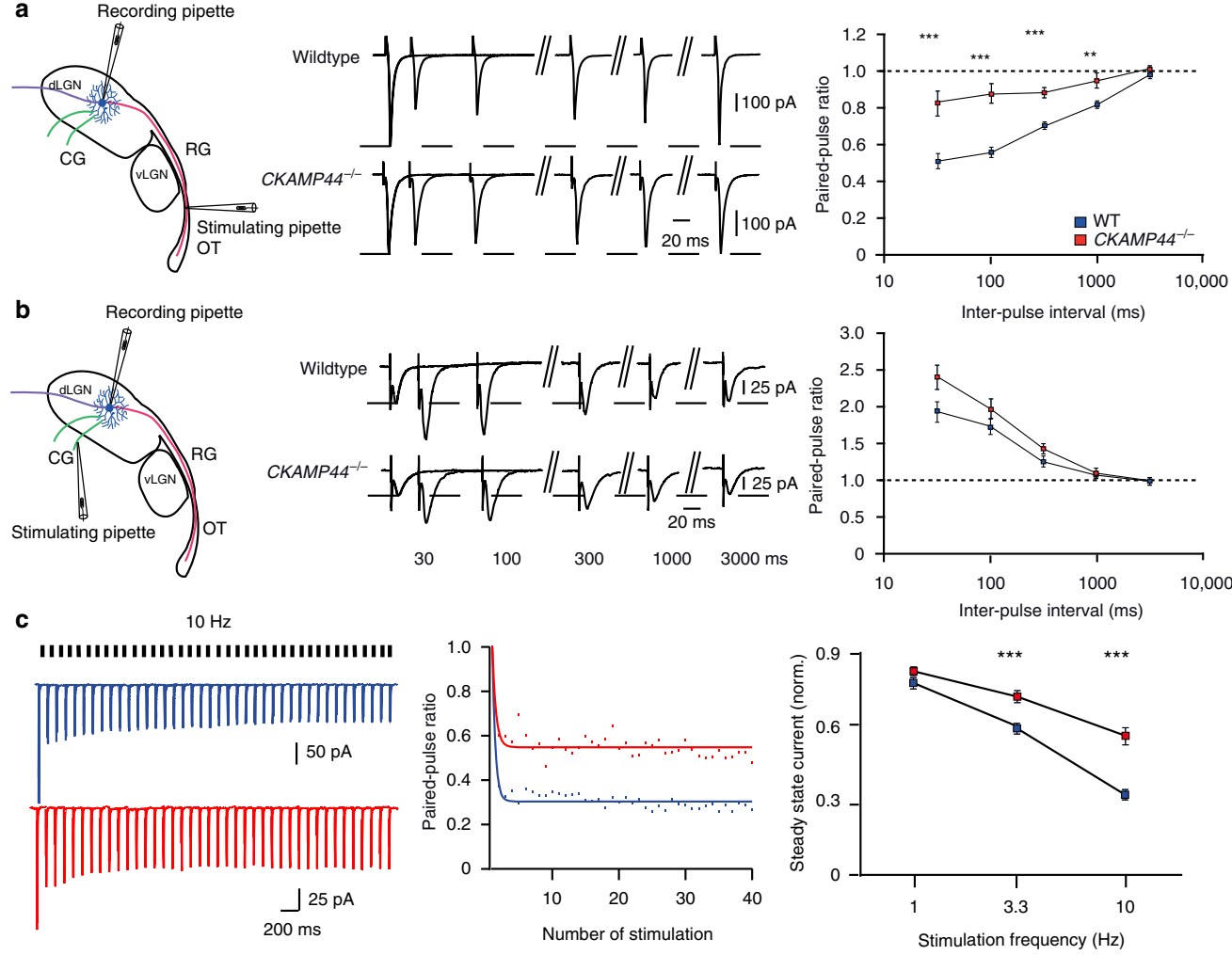

**Fig. 2** CKAMP44 modulates synaptic short-term plasticity in retinogeniculate synapses. **a** The PPR is increased in retinogeniculate synapses of *CKAMP44⁻/⁻* mice (n = 20 for relay neurons of wildtype and 18 for relay neurons of *CKAMP44⁻/⁻* mice, mean ± SEM, 300 ms inter-stimulus interval: *t*-test; 10–1000 ms inter-stimulus interval: Mann–Whitney test). **b** There is no significant change in the PPR in corticogeniculate synapses of *CKAMP44⁻/⁻* mice (n = 25 for relay neurons of wildtype and 22 for relay neurons of *CKAMP44⁻/⁻* mice, mean ± SEM, 300 ms and 3000 ms inter-stimulus interval: *t*-test, 10, 30, 100, and 1000 ms inter-stimulus interval: Mann–Whitney test). A schematic representation of RG and CG synapse activation with localization of stimulation pipettes is shown on the left of **a**, **b**. Example traces of pairs of currents that were evoked with two extracellular stimulations with different inter-pulse intervals (30, 100, 300, 1000, and 3000 ms) are shown in the middle. **c** The steady-state amplitude of synaptic currents is reduced in *CKAMP44⁻/⁻* mice when stimulating retinogeniculate synapses ×40 at 3.3 or 10 Hz (1 Hz; n = 25 for relay neurons of wildtype and 22 for relay neurons of *CKAMP44⁻/⁻* mice, 3.3 Hz; n = 31 for relay neurons of wildtype and 24 for relay neurons of *CKAMP44⁻/⁻* mice, 10 Hz; n = 33 for relay neurons of wildtype and 27 for relay neurons of *CKAMP44⁻/⁻* mice). The amplitudes were normalized to the amplitude of the first current (mean ± SEM, steady state EPSC amplitudes with 1 Hz stimulation: Mann–Whitney test, other data: *t*-tests). Example trace of 40 EPSCs evoked by stimulation of retinogeniculate synapses with 10 Hz are shown on the left and a double-exponential fit of the current amplitudes of the same data in the middle. *$p < 0.05$, **$p < 0.01$, ***$p < 0.001$

firing of RGCs before they fire with high frequency due to optimal visual stimuli. As expected from the voltage clamp experiments shown in Fig. 2c, EPSP amplitudes attenuated more in retinogeniculate synapses of wildtype than in those of *CKAMP44⁻/⁻* mice (Fig. 3c–e and Supplementary Table 14). The amplitude attenuation explains a reduced spike probability during the high frequency stimulus train (Fig. 3c–e and Supplementary Table 11). However, spike probabilities and firing rates were still higher in retinogeniculate synapses of *CKAMP44⁻/⁻* than in those of wildtype mice (Fig. 3c–e, Supplementary Tables 11 and 12). Of note, firing probability and firing rate were higher in *CKAMP44⁻/⁻* mice although stimulation strength was adjusted such that the first EPSP amplitude was 42–46% smaller in relay neurons of *CKAMP44⁻/⁻* mice than in those of wildtype mice (Supplementary Table 13). Current clamp experiments thus

suggest that deletion of CKAMP44 increases EPSP amplitudes and firing probability of dLGN relay neurons despite the fact that it reduces synaptic AMPAR number. The most parsimonious explanation for these changes is a less pronounced short-term depression of synapses containing CKAMP44-less AMPARs that display fast recovery from desensitization.

The experiments so far suggested that the influence of CKAMP44 on the firing rate of relay neurons increases with spike number and firing frequency of RGCs. To test this hypothesis, we investigated relay neuron firing rates when stimulating the optic tract with different stimulus numbers (1–10) and frequencies (25, 37.5, and 50 Hz, stimulus train duration 200 ms). We again adjusted stimulation strength such that the first stimulation did not elicit an action potential and such that the 1st EPSP amplitude was about 50% smaller in

relay neurons of *CKAMP44*[−/−] mice than in those of wildtype mice. As expected, the difference in firing rates between genotypes increased with higher stimulation frequency. Firing rates were not different for short stimulus trains (1–4 stimuli). Deletion of CKAMP44 resulted in increased firing rates when the optic tract was stimulated more than four times. The difference in firing rates between genotypes increased with stimulation number (Fig. 4, Supplementary Tables 15–17). Retinogeniculate synapse strength is relatively strong[6] explaining a high action potential transfer rate between RGCs and

LGN relay neurons[2]. Thus, we investigated firing rates with stimulation intensity just sufficient to elicit an action potential during the first stimulus and observed a significant higher firing rate of relay neurons in *CKAMP44*[−/−] mice than in those of wildtype mice for long (4–10 stimuli), but not short stimulus trains (Fig. 4d, Supplementary Table 18). In conclusion, experiments from in vitro electrophysiological recordings suggested that CKAMP44 affects integration of excitatory inputs especially when RGCs fire at high frequency with many action potentials.

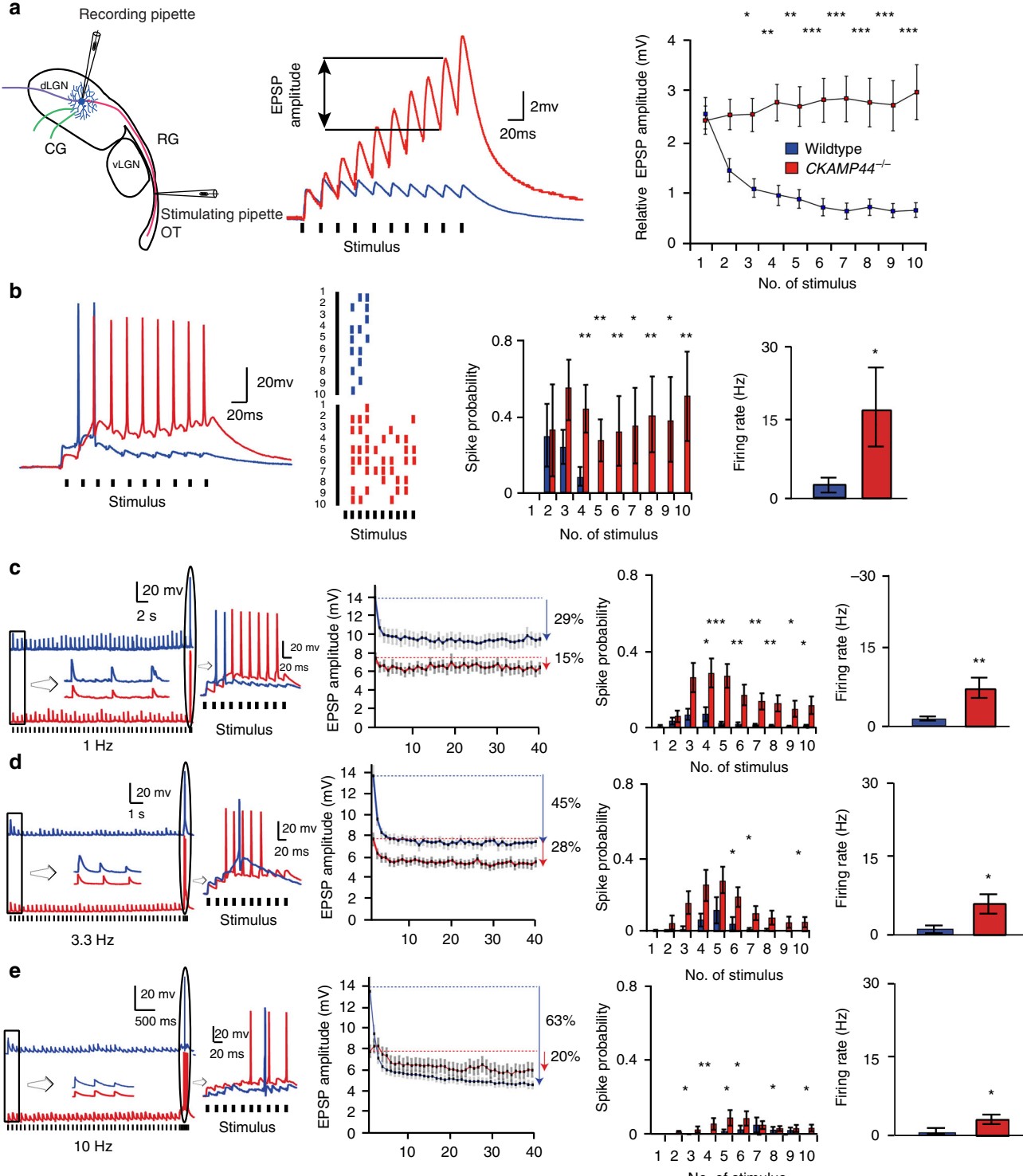

**CKAMP44 modulates relay neuron ON- and OFF-responses in vivo**. To investigate if CKAMP44 is relevant for dLGN neuron activity in vivo in response to natural visual stimuli that may elicit RGC activity at a wide range of frequencies and action potential numbers, we performed tetrode recordings from the dLGN of adult mice facing an LED monitor.

Before performing in vivo experiments, we analyzed in vitro if CKAMP44 influences AMPAR-mediated currents also in adult mice (and not only in P26–P33 mice, see above). Deletion of CKAMP44 indeed increased PPR of AMPAR-mediated currents (tested with 30 ms inter-stimulus interval) to a similar extent as in neurons of younger mice (Supplementary Fig. 7 and Supplementary Table 19), consistent with CKAMP44 expression in the LGN of adult mice as evidenced by in situ hybridization analysis (Fig. 1b).

To analyze ON- and OFF-responses of the relay neurons in vivo, we recorded from non-anaesthetized, head-fixed mice and applied whole-screen ON- and OFF-stimuli via a monitor (Fig. 5a). We recorded activity from a total of 1344 and 1144 cells in 6 wildtypes and 6 $CKAMP44^{-/-}$ mice, respectively. The number of recorded neurons per mouse was similar between genotypes (Supplementary Table 20). Also, the proportion of ON- and OFF-responding cells was not different between genotypes (Supplementary Table 20). However, ON and OFF peak rate responses and rate increases (in percentage of baseline firing rate) were higher in $CKAMP44^{-/-}$ than in wildtype mice, consistent with the increased spike probability in the in vitro experiments (Fig. 5, Supplementary Table 21; see Supplementary Fig. 8 for the same data plotted as mean ± SEM). Deletion of CKAMP44 did not affect mEPSC frequency and amplitude in RGCs and did not alter the responses to flash and pattern stimuli in the electroretinogram of adult mice either (Supplementary Fig. 9, Supplementary Tables 22 and 23), suggesting that relay neuron ON- and OFF-responses are increased in $CKAMP44^{-/-}$ mice not because of altered activity in the retina, but because of the absence of CKAMP44 in the dLGN.

## Discussion

Here we identified CKAMP44 as a crucial interacting protein of AMPARs in dLGN relay neurons. Using wildtype and $CKAMP44^{-/-}$ mice, we showed that this auxiliary subunit increases AMPAR-mediated current amplitudes and modulates gating of AMPARs. Importantly, by its modulation of the recovery from desensitization, CKAMP44 influences synaptic short-term plasticity and in consequence alters how excitatory input is integrated in the first relay station for visual information.

AMPAR function differs depending on the subunit composition[11,24,25] and on interacting proteins such as TARPs, cornichons, CKAMPs, and GSG1L[12–18,20,26]. Proteomic studies showed that each neuron type is endowed with a unique set of AMPAR subunits and interacting proteins[27], which explains the large variability in AMPAR gating properties in different neuron types[28]. Based on the long-lasting recovery from desensitization of AMPARs and the prominent contribution of desensitization to short-term plasticity in retinogeniculate synapses, we conjectured that CKAMP44 may be part of AMPAR complexes in this synapse. This is indeed the case as evidenced by in situ hybridization experiments and changes in AMPAR function in $CKAMP44^{-/-}$ mice. Electrophysiological experiments showed that CKAMP serves two functions in the dLGN: (i) it increases the surface expression of AMPARs and (ii) modulates their gating kinetics. The positive influence on the number of synaptic and extrasynaptic AMPARs is a function that CKAMP44 shares with several other auxiliary subunits such as TARPs and cornichons[29–31]. AMPAR-mediated current amplitudes are reduced by about 40% in $CKAMP44^{-/-}$ mice. This indicates that the remaining AMPARs interact with other auxiliary subunits since previous studies suggested that AMPARs may require the help of auxiliary subunits for an efficient trafficking to the cell surface[26,29,32–34]. In fact, TARP γ-2 and γ-4 are expressed in the dLGN and AMPAR-mediated current amplitudes are reduced in TARP γ-2$^{-/-}$ mice to a similar extent as in $CKAMP44^{-/-}$ mice[19].

Gating kinetics of AMPARs in dLGN neurons differ from those of AMPARs in other neurons and especially from those of homomeric or heteromeric AMPARs expressed in heterologous cells[11]. In particular, the slow recovery from desensitization ($\tau_{recovery} = 323$ ms; from this study), which plays an important role for short-term depression in retinogeniculate synapses, is considerably slower than that of homomeric or heteromeric AMPARs ($\tau_{recovery} = 6$–$67$ ms)[11]. This suggests that interaction with auxiliary subunits such as CKAMP39, CKAMP44, or GSG1L is responsible for the slow recovery from desensitization. Of these three proteins, only CKAMP44 displays high mRNA expression levels in the dLGN and the rate of recovery was indeed similar to that of heteromeric GluA1/GluA2-containing AMPARs in $CKAMP44^{-/-}$ mice. As TARP γ-2 is known to increase the rate of recovery from desensitization[12], it is possible that both auxiliary subunits are required for a balanced recovery rate.

In dLGN relay neurons of $CKAMP44^{-/-}$ mice, steady-state current amplitude and recovery from desensitization were increased. This may be explained by an increased stability of the desensitized state and thus a slowing of the exit from the desensitized conformation when AMPARs bind to CKAMP44. However, the desensitization time constant was not affected suggesting that the interaction of AMPARs with CKAMP44 does not alter the stability of the dimer interface between the S1–S2 domains of the AMPAR and thus does not modulate the rate of entry into the desensitized conformation. In contrast, the influence of TARP γ-2 on AMPAR gating properties indicated that this auxiliary subunit stabilizes the dimer interface between the S1

---

**Fig. 3** CKAMP44 decreases firing probability of dLGN neurons. **a** EPSP amplitudes increase more in neurons of $CKAMP44^{-/-}$ than in those of wildtype mice. EPSPs were evoked with a train of ten stimuli at 50 Hz (wildtype: $n = 18$, $CKAMP44^{-/-}$: $n = 25$, mean ± SEM, first and second EPSP amplitude: $t$-test, others: Mann–Whitney test). Example traces are shown in the middle, the quantification of the EPSP amplitudes on the right. **b** Spike probability during a $10 \times 50$ Hz stimulus is increased in $CKAMP44^{-/-}$ mice (wildtype: $n = 11$, $CKAMP44^{-/-}$: $n = 10$). Example traces and raster plots of action potentials during ten repetitions of the stimulus train are shown on the left. The spike probability (middle) for stimulus 4–10 and the firing rate during the entire stimulus train (right) is increased in $CKAMP44^{-/-}$ mice. Note that the stimulation intensity was adjusted such that the amplitude of the first EPSP was 48% smaller in neurons of $CKAMP44^{-/-}$ than in those of wildtype mice. (mean ± SEM, amplitude of responses 1–3: $t$-test, others: Mann–Whitney test). **c–e** Deletion of CKAMP44 increases spike probability and firing rate during a $10 \times 50$ Hz stimulus following a ×40 stimulation at 1 Hz (**c**, wildtype: $n = 27$, $CKAMP44^{-/-}$: $n = 33$), 3.3 Hz (**d**, wildtype: $n = 20$, $CKAMP44^{-/-}$: 32), and 10 Hz (**e**, wildtype: $n = 21$, $CKAMP44^{-/-}$: $n = 27$) (mean ± SEM, Mann–Whitney test). Example traces are shown on the left. The quantification (middle) shows that EPSP amplitudes during the ×40 stimulation train attenuate less and that firing probability is higher in relay neurons of $CKAMP44^{-/-}$ mice than in wildtype mice. The amplitudes of the first EPSPs are shown with dashed lines. Note that stimulation intensity was adjusted such that the amplitude of the first EPSP was 42–46% smaller in relay neurons of $CKAMP44^{-/-}$ than those in wildtype mice. Bar graphs (right) show the firing rate of relay neurons during the 200 ms stimulation. *$p < 0.05$, **$p < 0.01$, ***$p < 0.001$

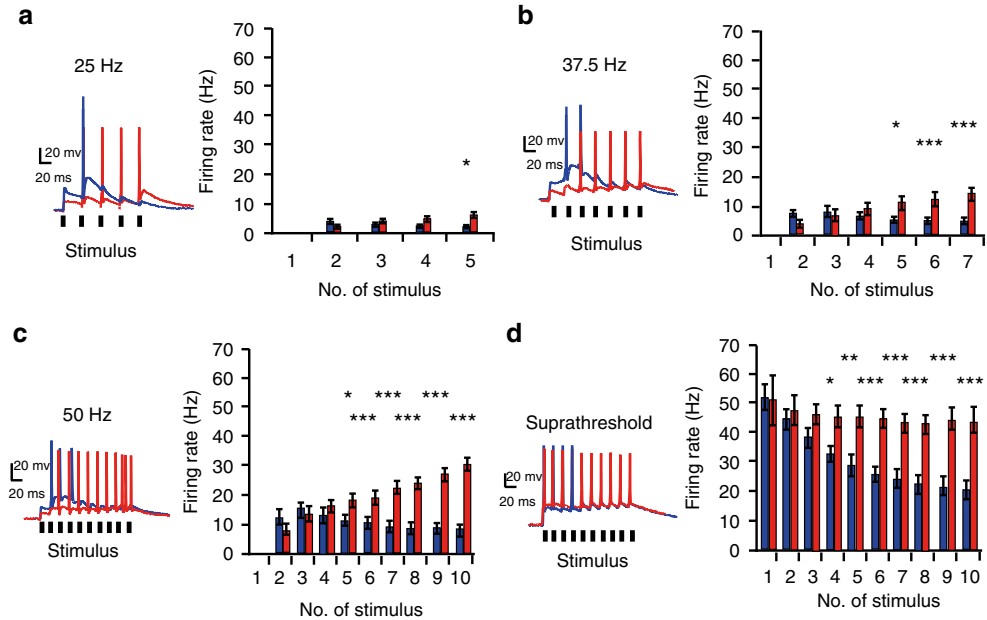

**Fig. 4** The influence of CKAMP44 on firing rate increases with optic tract stimulation number and frequency. **a–c** The difference in firing rates of relay neurons between genotypes increases with stimulation frequency and with the number of stimuli in the train. The optic tract was stimulated with 25 Hz (**a**, wildtype: $n = 50$, $CKAMP44^{-/-}$: $n = 67$), 37.5 Hz (**b**, wildtype: $n = 49$, $CKAMP44^{-/-}$: $n = 28$), and 50 Hz (**c**, wildtype: $n = 45$, $CKAMP44^{-/-}$: $n = 55$) and firing rates for different stimulus numbers were calculated by dividing the cumulative action potential number by the respective stimulation duration (mean ± SEM, Mann–Whitney test). **d** Firing rates are significantly higher in relay neurons of $CKAMP44^{-/-}$ mice than in those of wildtype mice when stimulating such that the first stimulus was just suprathreshold and elicited an action potential (wildtype: $n = 31$, $CKAMP44^{-/-}$: $n = 11$, mean ± SEM; responses to 3, 4, and 5 stimuli: Mann–Whitney test, other responses: t-test). $*p < 0.05$, $**p < 0.01$, $***p < 0.001$

and S2 domains (thus reducing the rate and extent of desensitization) in addition to its influence on the conformation that favors an entry into the open conformation[12]. Interestingly, the influence of CKAMP44 depends on the cell type. Thus, CKAMP44 altered the rate of desensitization in addition to the rate of recovery from desensitization in dentate gyrus granule cells[26]. The different modulation exerted by CKAMP44 may be due to the fact that AMPAR complex composition is not the same in the two cell types (e.g., GluA1/2+TARP y-8 in dentate gyrus granule cells and GluA1/4+TARP y-2 in dLGN relay neurons[35,36]).

AMPAR desensitization contributes substantially to synaptic short-term depression in retinogeniculate synapses[7,37]. Pharmacological block of AMPAR desensitization with CTZ decreases synaptic short-term depression in retinogeniculate synapses[7,10] to a similar extent as the genetic deletion of CKAMP44, suggesting that the postsynaptic influence of AMPAR desensitization on short-term plasticity depends mainly on the presence of this auxiliary subunit. Budisantoso and colleagues had tested if synaptic short-term depression is influenced by the expression of GluA1, a subunit that also confers slow recovery rates on AMPAR complexes. Short-term depression was indeed decreased in GluA1[-/-] mice[10], although to a much smaller extent than in $CKAMP44^{-/-}$ mice. The pronounced and long-lasting AMPAR desensitization depends also on the morphological structure and the high release probability of retinogeniculate synapses, which allows glutamate to spill over to non-active neighboring synapses and to dwell in the synaptic cleft for extended time periods[10].

Release probability and glutamate transporter activity increase with temperature[38,39]. A faster glutamate reuptake decreases the level of glutamate spillover at physiological temperature when compared to room temperature[38]. High release probability should increase the influence of CKAMP44 on short-term plasticity, whereas low levels of spillover glutamate should have the opposite effect. In fact, CKAMP44 influences short-term plasticity when

recordings are performed at room temperature (in retinogeniculate synapses; this study) and at physiological temperature (in perforant path-to-granule cell synapses[26]). Low release probability of corticogeniculate synapses[22] explain why AMPAR desensitization does not significantly contribute to short-term plasticity in this synapse. Consistently, PPRs were not altered in corticogeniculate synapses of $CKAMP44^{-/-}$ mice although the auxiliary subunit is part of AMPAR complexes also in this synapse.

In situ hybridization analyses (this study and Allen Brain Atlas) show a strong CKAMP44 mRNA signal in several thalamic nuclei including the medial geniculate nucleus, the ventral posterior nucleus and the mediodorsal thalamus. Interestingly, pronounced short-term depression has been described for excitatory inputs to neurons of these thalamic nuclei[40,41]. In addition, axons from the paleocortical piriform cortex form giant synapses with multiple synaptic contacts and strong single fiber input onto neurons from the mediodorsal thalamus[42]. These morphological characteristics may promote AMPAR desensitization due to glutamate spillover and thus could contribute to short-term depression similar to what has been reported for retinogeniculate synapses[10]. It is thus conceivable that CKAMP44 modulates short-term plasticity in several thalamic nuclei.

Transmission of visual information is most effective when RGCs fire at high frequency. Spike transmission is significantly higher for a second action potential than for a first if inter-event intervals are shorter than 30 ms[2]. This indicates that synaptic short-term plasticity in retinogeniculate synapses contributes to the processing of visual information. Voltage-clamp data from this and other studies showed that absolute EPSP amplitudes indeed increase due to summation during high frequency stimulation although EPSC and relative EPSP amplitudes decreases[3]. We showed that AMPAR desensitization reduces retinogeniculate synapse strength, which should theoretically act as a brake that prevents overactivation of dLGN neurons

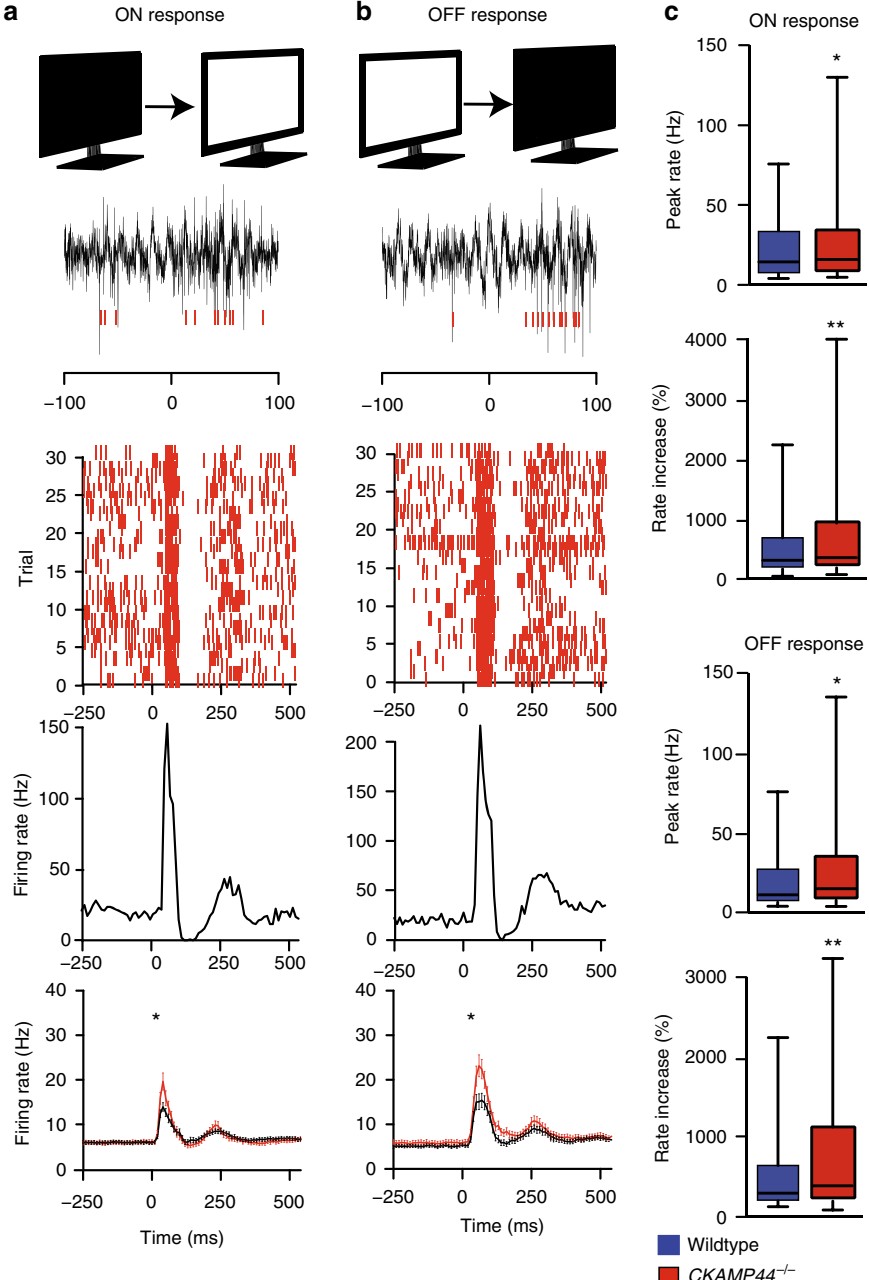

**Fig. 5** CKAMP44 decreases peak firing rate of ON and OFF-responses in vivo. **a** ON-response amplitude is significantly higher in $CKAMP44^{-/-}$ than in wildtype mice. ON-responses were observed briefly after increasing the light intensity of the monitor ($n = 542$ dLGN neurons in 6 wildtype and 430 dLGN neurons in 6 $CKAMP44^{-/-}$ mice; mean ± SEM). **b** Deletion of CKAMP44 increases also the OFF-response amplitude (mean ± SEM, Mann–Whitney test) ($n = 309$ dLGN neurons in 6 wildtype and 285 dLGN neurons in 6 $CKAMP44^{-/-}$ mice). **a**, **b** Raw traces of tetrode recordings are shown on the top, raster and peristimulus time histograms in the middle and average ON- and OFF-responses at the bottom. Red bars indicate spikes of dLGN neurons. **c** Quantification of the peak firing rate and the rate increase for ON- and OFF-responses (box-and-whisker plots showing the median, IQR and 5–95% range, Mann–Whitney test). *$p < 0.05$, **$p < 0.01$, ***$p < 0.001$

especially when RGCs fire at high frequency and with many action potentials. Indeed, dynamic-clamp experiments suggested that short-term depression reduces the likelihood that a dLGN neuron fires when activated with naturally occurring stimulation patterns[3]. Here we demonstrate that altering short-term depression by genetic deletion of CKAMP44 influences integration of excitatory inputs and spike probability. Thus, EPSP amplitudes and spike probabilities are significantly higher in relay neurons of $CKAMP44^{-/-}$ mice than in those of wildtype mice if retinogeniculate synapses are activated with a train of stimuli and the

difference in firing rates increases with the frequency and number of stimuli. This indicates that CKAMP44 prevents overactivation of those dLGN neurons that receive particularly strong inputs from RGCs (i.e., with high frequency and many action potentials). Importantly, the responses of dLGN neurons were increased in $CKAMP44^{-/-}$ mice despite a reduced number of synaptic AMPARs. In other words, the reduction in synaptic strength in $CKAMP44^{-/-}$ mice leads to an underestimation of the impact of short-term plasticity on integration of excitatory inputs. Moreover, the increase in firing rate of dLGN neurons

of CKAMP44$^{-/-}$ mice is observed with different low-frequency activities of retinogeniculate synapses preceding the high-frequency activation. This is important since non-transmitted low-frequency activity may influence synaptic strength and the extent to which short-term-depression influences transmission rate of subsequent high-frequency activity.

The increased firing rate of ON- and OFF-responses in non-anesthetized CKAMP44$^{-/-}$ mice shows that the increased responses of neurons in the acute brain slice experiments are not an artifact of the in vitro preparation and recording conditions such as room-temperature, which affects release probability, glutamate reuptake, spillover glutamate, and AMPAR gating properties[38,39,43], or the artificial stimulation protocol. Mouse ON and OFF RGCs respond in vivo with peak firing rates of 20 Hz to the presentation of full-field ON- or OFF-stimuli, respectively. In addition, RGCs have been described to fire with "background" rates of ~1–10 Hz when the intensity of the monitor is at average luminance (i.e., gray monitor)[44]. Thus, RGC firing frequencies are in the range of frequencies with which we stimulated the optic tract. A 25 Hz stimulus, thus similar to the peak firing frequency of RGCs during the presentation of full-field ON- or OFF-stimuli, activated dLGN neurons with a relative small difference in firing rates between genotypes, comparable to the difference in peak firing rates of dLGN neurons in vivo when CKAMP44$^{-/-}$ and wildtype are stimulated with full-field ON- or OFF-stimuli. Presentation of high or low luminance spots with optimal spot size elicits responses of RGCs with 6–10 spikes per coding event and peak firing rates of above 50 Hz, thus considerably higher than full-field ON- or OFF-stimuli[44,45]. RGC peak firing frequencies can even go up to 500 Hz during presentation of natural visual stimuli[45]. The in vitro analyses showed that the influence of CKAMP44 on relay neuron firing rates increases with stimulation frequency and number of stimuli, suggesting that CKAMP44 plays a bigger role for input integration when mice are presented with visual stimuli that elicit higher RGC responses than the simple full-field ON- or OFF-stimuli applied in the present study.

Approximately 30% of RGC spikes result in firing of dLGN neurons and those that are relayed carry more visual information than non-transmitted spikes[1,2]. CKAMP44 reduces responses of dLGN neurons to high-frequency input and at the same time augments synaptic strength by increasing synaptic AMPAR numbers. Thus, it is likely that CKAMP44 not only prevents overactivation of dLGN neurons, but also modulates the information content that is relayed to the visual cortex by influencing RGC action potential transmission.

## Methods

**In situ hybridization with oligoprobe**. In situ hybridization with an oligoprobe was performed on horizontal brain sections from C57BL/6 mice. Brains were removed and frozen on dry ice. In total, 14 µm sections were cut with a cryostat and slide-mounted, fixed for 5 min in 4% paraformaldehyde, rinsed in PBS and dehydrated in a series of ascending ethanol concentrations. Sections were stored in 100% ethanol at 4 °C until use. In situ hybridization with a 35S-labeled antisense 45-mer oligodeoxyribonucleotide probe (c24.is3, 5′-GCA TGA CCC AGG AAA AGC ATG ACT CCT TAT GAG TAG GTC TGT GGT-3′) to the 3′ UTR of CKAMP44 was performed as described by von Engelhardt et al.[18].

**In situ hybridization with riboprobe**. To prepare the RNA probe, a ~1.5 kb PCR product was amplified from mouse brain cDNA using a primer pair (forward primer 5′-GGTCAATGATGACTTCTACGCC-3′; reverse primer 5′-ACCTCTGACATTGTAGCAGCTC-3′) covering the exon 4 and 3′ UTR region of the mouse Ckamp44 gene (NM_028277). The PCR fragment was cloned into pCRII-TOPO vector (Invitrogen, Carlsbad, CA). Digoxigenin (DIG)-labeled RNA probe was synthesized using T7 RNA polymerase and the DIG RNA labeling kit (Roche) according to manufacturer's guidelines. For in situ hybridization, brains from 4-week-old C57BL/6 mice were removed and rinsed once in Diethylpyrocarbonate treated PBS (DEPC-PBS). Brains were immediately frozen on dry ice.

Horizontal sections (20 µm thickness) were cut on a cryostat (Leica Microsystems, Germany) and mounted onto Superfrost plus slides (Thermo Scientific, Braunschweig, Germany). Sections were treated with 4% PFA for 20 min and acetylated with 100-mM triethanolamine (pH8.0) in 0.25% acetic anhydride for 10 min with stirring. Sections were then rinsed with DEPC-PBS and treated with 1 µg/ml Proteinase K (Roche) for 10 min. Pre-hybridization was performed in solution containing 50% formamide, 5× SSC, 0.1% Tween 20, and 0.3 mg/ml Yeast tRNA, 0.1 mg/ml Heparin, 1× Denhardt's Solution and 5 mM EDTA in DEPC-H2O (Sigma) at 63.5 °C for 2 h. Hybridization was performed in the same solution with 0.1–0.2 µg/ml of denatured DIG-labeled RNA probe at 63.5 °C overnight. After post-hybridization washing steps, the hybridized probes were reacted with alkaline phosphatase-conjugated anti-DIG antibody (1:2000; Roche). Signals were developed in alkaline phosphatase buffer containing nitroblue tetrazolium chloride (NBT) and 5-bromo-4-chloro-3-indolyl-phosphate (BCIP).

**dLGN slice preparation**. Acute brain slices for electrophysiological analysis of dLGN neurons were prepared as described by Turner and Salt[46]. Briefly, the brain was quickly removed from anesthetized P26-P33 and adult C57BL/6 mice and then immersed in oxygenated 4 °C saline solution containing (in mM): 87 NaCl, 2.5 KCl, 37.5 choline chloride, 25 NaHCO$_3$, 1.25 NaH$_2$PO$_4$, 25 glucose, 0.5 CaCl$_2$, and 7 MgCl$_2$. A total of 250–300 µm thick brain sections were cut on a vibratome (Sigmann HR2) with a cutting angle described by Turner and Salt[46]: The two hemispheres were separated with a 3–5° angle to the sagittal plane and a 10–25° angle outwards in the mediolateral plane. The medial aspect of each hemisphere was then glued onto the cutting stage. Slices were kept in oxygenated saline at 34 °C to recover for 30 min and then transferred into recording ACSF containing (in mM): 125 NaCl, 25 NaHCO3, 1.25 NaH$_2$PO$_4$, 2.5 KCl, 2 CaCl$_2$, 1 MgCl$_2$, 25 glucose at 34 °C to recover for another 30 min before electrophysiological recordings.

**In vitro electrophysiology**. In vitro recordings were performed at room-temperature using pipettes pulled from borosilicate glass capillaries with a resistance of 3–5 MΩ (whole cell experiments) or 5–7 MΩ (nucleated patch experiments). Unless mentioned, pipettes were filled with Cs$^+$ containing solution for voltage clamp experiments (in mM): 35 Cs-gluconate, 100 CsCl, 10 HEPES, 10 EGTA, and 0.1 D-600; (pH 7.3, adjusted with CsOH). To compare short-term plasticity in current and voltage clamp recordings and to exclude that differences in PPR between genotypes were augmented by Cs$^+$ blown onto the slice, we performed voltage clamp PPR and 50 Hz train experiments also using a K$^+$ containing solution (in mM): 105 K-gluconate, 30 KCl, 10 HEPES, 10 phosphocre, 4 Mg-ATP, 0.3 GTP; (pH 7.3, adjusted with KOH). Liquid junction potentials were not corrected. Series resistance and input resistance were monitored at regular intervals by measuring peak and steady-state current amplitudes in response to small hyperpolarizing voltage steps. Slices were continuously perfused with ACSF containing (in mM): 25 NaCl, 25 NaHCO$_3$, 1.25 NaH$_2$PO$_4$, 2.5 KCl, 2 CaCl$_2$, 1 MgCl$_2$, and 25 glucose; bubbled with 95%O$_2$/5%CO$_2$ (pH 7.4).

Synaptic EPSCs were recorded by electrically stimulating either retinogeniculate fibers or corticogeniculate fibers. Retinogeniculate fibers were stimulated with an ACSF filled glass pipette that was placed into the optic tract. Corticogeniculate fibers were stimulated rostro-ventrally of the dLGN. AMPA/NMDA ratio was calculated from the amplitudes of AMPAR-mediated and NMDAR-mediated currents that were recorded at a holding potential of −70 mV and +40 mV, respectively. NMDAR-mediated current amplitudes were measured 25 ms after the start of the stimulus artifact. An aliquot of 10 µM SR 95531 hydrobromide was used to block GABA$_A$-receptors. Isolated AMPAR-mediated currents were recorded in the presence of 50 µM D-APV and 10 µM SR 95,531 hydrobromide to block NMDARs and GABA$_A$Rs, respectively. Kinetic properties of AMPARs were investigated in the presence and absence of 100 µM cyclothiazide (CTZ). Short-term plasticity was investigated with paired-pulse stimulation of retinogeniculate and corticogeniculate axons with 30, 100, 300, 1000, and 3000 ms inter-stimulus intervals. PPR of current amplitudes (EPSC$_2$/EPSC$_1$) was calculated from the means of 20 EPSC pairs. Retinogeniculate fibers were stimulated 40 times with 1 Hz, 3.3 Hz, or 10 Hz to investigate the steady-state depression of current amplitudes. EPSC amplitudes were normalized to the amplitude of the first EPSC (EPSC$_n$/EPSC$_1$). The normalized EPSC amplitudes were fitted with an exponential curve to estimate the steady state EPSC amplitude.

EPSPs, spike probabilities and firing rates were analyzed in current clamp-mode while stimulating retinogeniculate fibers. Different stimulation protocols (10 stimuli at 50 Hz, 7 stimuli at 37.5 Hz, 5 stimuli at 25 Hz, and 40 stimuli at 1 Hz, 3.3 Hz, or 10 Hz followed by 10 stimuli 50 Hz following) were applied. Stimulation intensities were adjusted to elicit action potentials (just above action potentials threshold stimulus intensity) or EPSPs with defined amplitudes. Each stimulation protocol was applied ten times and averages were used to analyze EPSP amplitudes. EPSP amplitudes were calculated as the difference between the potential preceding the EPSP and the peak of the EPSP (Fig. 3). Spike probability was calculated for each stimulus during the different stimulus trains. Firing rates for each stimulus numbers (shown in Fig. 4) were calculated by dividing the cumulative action

potential number by the stimulation duration at the given stimulus number (e.g., for stimulus 5 at 50 Hz: × action potentials/0.1 s).

Nucleated patch experiments were performed as described[47] using theta glass tubing mounted onto a piezo translator. Application pipettes were tested by perfusing solutions with different salt concentrations through the two barrels onto open patch pipettes and recording current changes with 1 and 100 ms transitions of the application pipette. Only application pipettes with 20–80% rise times below 120 μs and with a reasonable symmetrical on- and offset were used. The application solution contained (in mM): 135 NaCl, 10 HEPES, 5.4 KCl, 1.8 CaCl$_2$, 1 MgCl$_2$, 5 glucose (pH 7.2). AMPAR-mediated currents were evoked by applying 1 and 100 ms glutamate (1 mM) pulses onto nucleated patch patches of relay cells.

Relay neurons were visually identified using an upright microscope equipped with infrared-differential interference contrast and standard epifluorescence. To analyze decay and desensitization rates, AMPAR-mediated currents were fitted with single- or double-exponential functions. Double-exponential fits of EPSCs were used when the chi-square error of the fit was smaller than that for the single-exponential fit. In that case, a weighted time constant $\tau_w$ was calculated as $\tau_w = (\tau_f \times a_f) + (\tau_s \times a_s)$, where $a_f$ and $a_s$ are the relative amplitudes of the fast ($\tau_f$) and slow ($\tau_s$) exponential components.

**Surgical procedure for in vivo recordings.** In vivo experiments were performed with 3–4-month-old female C57BL/6 mice (6 wildtype and 6 $CKAMP44^{-/-}$ mice) and were approved by the Governmental Supervisory Panel on Animal Experiments of Baden-Wuerttemberg in Karlsruhe (35-9185.81/G-14/15). Animals were singly housed and kept on a 12-h light–dark schedule. Recordings took place during the light phase. Microdrives carrying eight movable tetrodes were prepared. Tetrodes were made of 12-μm-diameter tungsten wires (H-Formvar insulation with Butyral bond coat; California Fine Wire Company) and were gold-plated to reduce impedance to 300–400 kΩ.

Mice were anesthetized with isoflurane (3% induction, 1.5% maintenance) and placed into a stereotaxic apparatus. Eyes were covered with Bepanthen (Bayer, Germany) during surgery and body temperature was kept at 37 °C via a feedback controlled heating pad. The skull was exposed and four anchor screws were inserted into the skull. Two screws located above the cerebellum served as ground and reference signals. The following coordinates were used (medial–lateral: 2.0 mm from sagittal suture, anterior–posterior: −2.0 mm from bregma). Tetrodes were inserted to a depth of 2.5 mm below the brain surface, corresponding to the dorsal border of the dLGN. The microdrive was fixed to the skull with dental cement. Mice were given 1–2 weeks to recover after surgery.

Analgesics (Carprofen, 5 mg/kg s.c.) were administrated at the end of the surgery. A long-lasting analgesic (Buprenorphin, 0.1 mg/kg and Carprofen, 5 mg/kg s.c.) was administered daily for 3 days after the surgery.

**Visual stimulation.** After a 1–2 weeks' recovery period, the animal was habituated to the head-fixation procedure in the recording setup. A 24-inch monitor (X2411Z, BenQ) was placed 25 cm away from the head-fixed mouse with a viewing angle of ~45° on left and right side, ~37° above and below. Visual stimuli were displayed on a gamma-corrected LED monitor with a refresh rate of 140 Hz. Visual stimuli were controlled through custom-made software based on Visual C++ and OpenGL. The monitor luminance changed every 2 s. The monitor luminance alternated every 2 s between 100 and 0% of the maximal intensity.

**Data acquisition.** Tetrodes were lowered into the dLGN over several days. Correct position of the tetrodes in the dLGN was identified by strong increases in unit activity in response to whole screen ON- and OFF-stimulation. After each recording session, tetrodes were lowered deeper into the dLGN by 25–50 μm to collect data from additional neurons. Subsequent recording sessions were performed 6–24 h after changing the tetrode position. The signal was amplified, and digitized at 20 kHz (RHD2000-Series, Amplifier Evaluation System, Intan Technologies, analog bandwidth 0.09 Hz–7.6 kHz). The brain of each mouse was collected after completing the experiments and correct tetrode locations were confirmed by histological analysis.

**Spike detection and sorting.** Spike detection and sorting To detect action potentials, a bandpass filter (0.8–5 kHz) was applied to the raw signal and the root mean square was calculated in 0.3 ms time windows. Time windows exceeding the mean root mean square by more than 5 SDs were considered as action potentials [48]. Spike waveforms were extracted from the four wires of the tetrode and principal component analysis was applied to the waveforms of each wire separately. The first three principal components of each wire were concatenated and served as spike features. Spike clusters were automatically generated with KlustaKwik (https://github.com/klusta-team/klustakwik)[49] and were manually refined with a graphical user interface. Clusters without a clear refractory period were not analyzed further. Spike-time autocorrelations were constructed by considering every spike in turn as a reference spike to all other spikes. The time differences between the reference spike and all other spikes were calculated and compiled into a

histogram (1 ms bins). The spike count in each bin of the histogram was divided by the total number of spikes to obtain a firing probability.

ON and OFF cells were identified from their peristimulus time histogram (−1000 to 1000 ms, 10 ms per bin) for either positive (ON) or negative (OFF) luminance changes. The response of a neuron was estimated from its peak rate between time 0 and 100 ms. The mean and standard deviation of the baseline activity was calculated from time −1000 to 0 ms. Responses with a peak response that was more than three standard deviations above the baseline mean were considered as ON- or OFF-responses. Neurons with fewer than 400 spikes during the 10–30 min recording time and a peak firing rate below 5 Hz in both ON and OFF crosscorrelations were not considered for the ON- and OFF-response analysis. In addition, cells recorded on tetrodes with no ON- and OFF-cell during all recording sessions (4 out of 93 tetrodes) were not included for the estimation of the proportion of neurons with ON- or OFF-responses, as we could not exclude that these tedrodes were located outside of the LGN.

**Retina isolation and electrophysiology.** Retinas were prepared from adult C57BL/6 mice (age 3–4 months) as described by Schmidt and Kofuji[50]. Briefly, after cervical dislocation, eyes were enucleated. Cornea, iris, lens, and vitreous were then removed from the eye. Retinas were detached from the retinal pigment epithelium layer and immersed in a Petri dish filled with oxygenated extracellular solution containing: 8.8 g/l Ames' medium and 23 mM NaHCO$_3$. To digest the remaining vitreous and allow for easier access to the ganglion cell layer, retinas were kept for 15 min with gentle shaking in a solution containing: 8.8 g/l Ames' medium, 23 mM NaHCO$_3$, 240 U/ml Collagenase (Worthington Chemicals) and 6000 U/ml Hyaluronidase (Worthington Chemicals).

Retinas were then transferred into the recording chamber with the concave side (retinal ganglion cell layer) facing upwards and continuously perfused with oxygenated extracellular solution containing: 8.8 g/l Ames' medium and 23 mM NaHCO$_3$, 1 μM tetrodotoxin to block spontaneous activity, 50 μM D-APV to block NMDARs, 10 μM SR 95,531 hydrobromide to block GABA$_A$Rs and 10 μM strychnine to block GlyRs. mEPSCs were recorded at a holding potential of −70 mV. The pipette solution for mEPSC recordings contained (in mM): 120 Cs-gluconate, 10 CsCl, 10 HEPES, 0.2 EGTA, 10 phosphocre, 8 NaCl, 2 Mg-ATP, 0.3 GTP; (pH 7.3, adjusted with CsOH).

**Electroretinogram.** Electroretinogram (ERG) recordings were obtained from 6 wildtype and 6 $CKAMP44^{-/-}$ mice, age 3–4 months. Experiments were approved by the Governmental Supervisory Panel on Animal Experiments of Baden-Wuerttemberg in Karlsruhe (35-9185.81/G-38/16). ERG experiments were performed as described by Mura et al.[51]. Briefly, animals were kept under 12-hour dark/light cycles and dark-adapted 6–12 h before recording. Dark-adapted mice were anesthetized by intraperitoneal injection of ketamine (150 mg/kg) and xylazine (10 mg/kg). A contact lens electrode with gold contact (LKC Technologies, Gaithersburg, Maryland) was placed on the corneal surface. The electrical contact was increased and eye desiccation prevented by application of Liquifilm O.K. (Allergan, Westport, Ireland). A heating pad controlled by a rectal temperature probe maintained mouse body temperature at 37 °C. Reference and ground needle electrodes were placed subcutaneously in the neck and tail, respectively.

For flash ERG recordings, the pupil of one eye was dilated with 1% atropine sulfate. The other eye was protected with Bepanthen (Bayer, Germany). Full-field ERGs were recorded by placing the mouse into a Ganzfeld dome equipped with an LED whole-field stimulator. Light flashes were presented at 0 dB intensity (2.5 cd/m$^2$ luminance) and 2 Hz using a UTAS BigShot (LKC Technologies). One-hundred sweeps were averaged for each ERG recording, amplified 1000 times and band-pass filtered between 0.3 and 500 Hz before digitization (sample rate 1000 Hz).

For pattern ERG (pERG) recordings, a 17-inch CRT monitor (Dell) was positioned 20 cm in front of the eye, and centered in a position ~40° medially from the pupil axis. In order to achieve maximum focus of the pattern stimulation, pupils were not dilated. Vertical square wave grating stimulations (66% contrast, temporal frequency of 2 Hz) were presented at different spatial frequencies (0.025, 0.05, 0.1, and 0.2 cpd), and noise levels were measured with the monitor turned off. The second harmonic amplitude of the pERG (i.e., the harmonic that had a frequency twice that of the stimulus) was analyzed from the Fourier transform of averages of 1000 sweeps.

**Statistics.** Data are presented as mean ± standard error of the mean (SEM). Data that were not normally distributed are presented as box-and-whisker Tukey plots showing the median, interquartile range (IQR) and 1.5 times the IQR. Statistical differences between groups were examined by two-tailed Student's $t$ test when the values showed a normal distribution. For non-Gaussian distributed data, we used two-tailed Mann–Whitney. Normality of data distribution was tested by Shapiro–Wilk normality test and equal variance by Levene Median test. In vitro analysis was performed using sigma plot (version 11.0). In vivo analysis was performed with R. $P$ values < 0.05 were considered statistically significant (*$p < 0.05$, **$p < 0.01$, *** $p < 0.001$).

**Data availability**. Data supporting the findings of this study are available within the article and its Methods section, and from the corresponding authors upon reasonable request.

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

## Acknowledgements

This work has been funded by the German Research Foundation (DFG) within the Collaborative Research Center (SFB) 1134 "Functional Ensembles" (J.v.E. and X.C.) and the Research Grant EN948/1-2 (J.v.E.). We are grateful to Paul Farrow and Richard Fairless for their technical assistance.

## Author contributions

X.C. performed in vitro and in vivo recordings. X.C. and K.A. analyzed in vivo experimental data. M.A. performed in situ hybridization. T.G. provided visual stimulation protocols. X.C., M.A., T.G., K.A., and J.v.E. conceived the experiments and wrote the manuscript.

## Additional information

**Competing interests:** The authors declare no competing financial interests.

