## [Peer Review File · Nature Communications]

Reviewers' comments:

Reviewer #1 (Remarks to the Author):

The authors assess the role of CKAMP44, an AMPAR auxiliary subunit, in excitatory synaptic processing by cells of the dorsal lateral geniculate (dLGN), focusing mainly on the retinogeniculate pathway. First, they show that CKAMP44 is expressed in the dLGN. They then go on to examine the functional roles of the protein in the dLGN by comparing the neurophysiology of dLGN relay neurons from control mice and from mice with genetic deletion of CKAMP44 (CKAMP44^{-/-}). There were essentially no effects on intrinsic membrane properties but there were a number of effects on AMPAR-mediated responses. AMPA responses were generally smaller in the CKAMP44^{-/-} cells, consistent with a role in AMPA receptor trafficking. There were also effects on AMPAR "gating" in retinogeniculate synapses, supporting roles for CKAMP44 in AMPAR desensitization and short-term synaptic depression. Consistent with these roles in synaptic dynamics, the authors provide evidence that deletion of CKAMP44 enhances synaptic integration and evoked firing of dLGN cells, including visually-evoked firing in awake mice.

This study is thorough and the experiments appear to be nicely done. The manuscript extensively describes the role of CKAMP44 on AMPA responses in dLGN neurons at multiple levels of analysis. Yet, some of the experiments appear to at least partially contradict one another. Furthermore some of the analysis and writing need to be corrected so the main ideas in the manuscript are clear to the reader.

Major issues:

1. The voltage clamp experiments in Figure 2 show that the retinogeniculate synapse is depressing, as expected from decades of previous studies. CKAMP44 deletion was found to significantly weaken the short term depression, although some clear depression remained. Surprisingly, the current-clamp data in Figure 3A shows much weaker depression than would be predicted by the voltage-clamp recordings. In fact, the EPSPs from the CKAMP44 KO mice actually facilitated. The differences between the voltage-clamp and current-clamp results need to be addressed and reconciled. A major methodological difference between the voltage-clamp and current-clamp recordings involved the internal recording solutions, with cesium being used for voltage-clamp and potassium for current-clamp. It is possible that cesium, ejected onto the retinogeniculate terminals during the patching procedure, artificially enhanced short-term depression of the retinogeniculate synapses. I have personally observed such an effect of cesium in another pathway, and it required nearly an hour of rinsing for the effects of the cesium solution to dissipate following patching (and for short-term dynamics to normalize). I suggest that the authors test the excitatory synaptic currents (in voltage clamp) using a potassium-based internal solution to determine whether the differences in short-term dynamics observed in voltage-clamp and current-clamp have an underlying "neurobiological explanation" (e.g., activation of intrinsic conductances in current-clamp, leading to apparent "boosting" of synaptic responses during trains), or are instead due to a technical artefact.

2. There is conflation of facilitation and summation. The authors often seem to be making points about summation of EPSPs but they use the term "facilitation". Facilitation and summation are two separate phenomena. Both of these dynamic aspects of the responses could be important for a clear understanding of the roles of CKAMP44. The Methods and Results should clarify which of the phenomena are being measured and exactly how it is done.

a. In the CKAMP44 KO trace shown in Figure 3a, there are two distinct phenomena occurring: facilitation and summation. The data analysis should account for these separately, and should be clearly explained. In order to address facilitation/depression of EPSP size, the authors should measure the peak amplitude of each EPSP relative to the baseline immediately preceding that particular EPSP. In contrast, measurements can be referenced to the baseline preceding the first stimulus in a train to assay summation of the EPSPs. The latter is what appears to be plotted in Figure 3A.

b. p.7: "Current clamp experiments thus suggest that deletion of CKAMP44 increases EPSP amplitudes and firing probability of dLGN relay neurons despite the fact that it reduces synaptic AMPAR number." I believe the word "amplitudes" should be replaced with "summation" here. The initial amplitudes are decreased rather than increased, and it is unclear to me whether or not the later amplitudes are actually greater than in the WT, although the summated response seems to be enhanced.

c. p.10: "This indicates that synaptic short-term facilitation of synaptic potentials in retinogeniculate synapses contributes to the processing of visual information." The normal retinogeniculate synapse has never, to my knowledge, been shown to facilitate. Instead, there can be summation at high rates.

3. There are a number of important mistakes in the writing, especially during summaries of the results. These mistakes could lead to real confusion. The authors should carefully revise the text with an eye to correcting such mistakes. Here are some examples:

a. p.3: "By its influence on short-term plasticity, CKAMP44 plays a role for input processing as evidenced by increased excitatory postsynaptic potential (EPSP) amplitudes and spike probability upon repetitive stimulation of the optic tract." This is confusing. I believe the authors are referring to effects of deletion of the protein, but the sentence is ambiguous and could be read to mean that the protein itself contributes to increased EPSPs upon repetitive stimulation.

b. p.6: "Voltage-clamp experiments showed that CKAMP44 increases short-term depression in retinogeniculate synapses by accelerating recovery from desensitization of AMPARs." The statement seems to be wrong; the evidence in the paper indicates that CKAMP44 slows recovery from desensitization of AMPARs rather than accelerating recovery.

c. p.6: "In stark contrast, current amplitudes increased steadily in synapses of CKAMP44^{-/-} mice (Fig. 3a and Table 6)." The authors are talking about synaptic potentials, not currents.

d. p.9: "Pharmacological block of AMPAR desensitization with CTZ increases synaptic short-term depression in retinogeniculate synapses to a similar extent as the genetic deletion of CKAMP44..." Both CTZ and deletion of the protein decrease depression rather than increasing it.

e. p.10: "Short-term depression was indeed increased in GluA1^{-/-} mice 10, although to a much smaller extent than in CKAMP44^{-/-} mice." Again, backward – depression was

decreased in both models, correct?

f. Figure 2c Legend: The steady-state amplitude is reduced (not increased, correct?) in the CKAMP44^{-/-} mouse?

g. Figure 3c-e Legend: There does not seem to be any explanation of the panels on the right.

h. Figure S4: There are no letters labeling the distinct panels.

i. The titles of the tables include references to figures, but the figure numbers seem to be wrong.

Minor issues:

1. The AMPA responses are reduced upon deletion of CKAMP44, but there seems to be no direct evidence in this paper regarding surface trafficking. The authors imply that there is direct evidence for surface trafficking (e.g., in the Abstract) when there doesn't seem to be much. Please tone it down.

2. In Figure 1g, there is no difference between WT and CKAMP44^{-/-} cells in terms of rates of desensitization. Instead, the differences are related to recovery from desensitization (Fig 1h) and steady-state currents. Please elaborate on the interpretation and importance of these subtleties with respect to mechanisms.

3. Regarding summation seen in Figure 3A, is it possible that mGluR receptors are being activated?

4. The Introduction and/or Discussion would benefit from elaborating on the potential role of CKAMP44 on other depressing synapses in parallel systems (e.g., medial lemniscus to ventral posterior synapse and inferior colliculus to medial geniculate synapse). Do these have CKAMP44, and can the results observed in retinogeniculate synapses be generalized to any other synapses?

5. For figure 2 and 3: Comment on use of naturally occurring firing patterns and frequencies. What stimulus frequencies correspond to typical RGC rates? Please elaborate on both "background" firing rates and on stimulus-evoked rates.

6. The Methods indicate that the in vitro recordings were performed at room temperature. Was this true even for the functional measures of synaptic integration and spiking? Were results different when recordings were done closer to physiological temperatures? This seems to be important considering the rather weak effects observed in vivo (Fig. 4). Please discuss.

7. Methods and/or Results section: elaborate on stimulus intensity. Are there any differences in the stimulus intensities used per genotype?

8. Abstract: Change "know" to "known"

9. Figure 1e, Contol should be Control

10. Results section for figure 4: Electroretinogram data in Figure s6 not s7.

11. Figure s2: 43% not 44%. Discrepancy between figure and text.

12. As far as I can tell, reference 36 does not actually describe the nucleated patch technique although it does seem to use the method. Please provide a reference that describes the technique.

Reviewer #2 (Remarks to the Author):

This electrophysiological study provides new insights regarding the regulation of AMPA receptors by the CKAMP44 auxiliary subunit. The authors specifically focus on retinogeniculate synapses because these are characterized by a robust short-term depression of AMPA receptor mediated currents and the mechanism is unknown. Through a series of rigorous studies the author demonstrate that CKAMP44 mediates this short term plasticity via its reducing the rate of AMPA receptor recovery from desensitization. Because the physiological roles for CKAMP44 and for AMPA receptor desensitization are poorly understood, this paper is of general interest.

The authors first show that CKAMP44 is expressed in dLGN neurons, where it controls the gating and amplitude of AMPA receptor mediated currents. These data are clear and convincing.

The authors next show that that CKAMP44 specifically modulates the paired pulse ratio in reticulogeniculate synapses but not in corticogeniculate synapses. Again, these experiments are well done and the results are clear.

The authors show that CKAMP44 decreases the firing probability of dLGN neurons. This is evidenced by many experiments such as the increase in EPSP amplitudes and the increase in spike probability during a 10x50 Hz stimulus in CKAMP44 KO mice.

Finally, the authors argue that CKAMP decreases peak firing rate in ON and OFF responses in vivo. These data are less convincing. Especially difficult to interpret are data in figure 4c, which quantifies the ON and OFF responses of dLGN neurons. The data are presented as median +/- IQR. The small differences between WT and KO and the very large IQRs make it hard to appreciate that the changes are significant. It would be important to see the individual data points and to have other statistics such as the S.E.M.

Reviewer #3 (Remarks to the Author):

This manuscript reports a functional role for the AMPAR auxiliary protein CKAMP44 in the modulation of AMPAR currents in the lateral geniculate nucleus (LGN). While TARPs have received the majority of attention in the relatively new field of AMPAR auxiliary subunits, we still know comparatively little about CKAMP44. It has been established in previous work by the senior author that the slowing of AMPAR recovery from desensitization by CKAMP44 can influence STP in the dentate gyrus. The current work reports a similar finding in the LGN, and goes further by presenting *in vivo* data suggesting that in a CKAMP44^{-/-} mouse line, visual processing is modified. This is the first report of an *in vivo* role for an auxiliary subunit that slows recovery from desensitization. As such it represents a good advance in our understanding. The data are of high quality, as we have come to expect from this lab, and the paper is well structured. However, there are issues which are a concern, and which clearly need to be addressed.

My major concern is that the *in vivo* changes, which are critical to the novelty of the study, appear to be of a very different magnitude to those detected in slice recordings. The data in Figures 3 and 4 seem poorly comparable and this does not appear to have been given due consideration in the study or the presentation. The obvious concern is whether the firing patterns elicited in Figure 3 are of relevance to the situation *in vivo*. Specifically, Figure 3 seems to demonstrate that wild-type cells should rarely be able to fire above 10 Hz despite direct stimulation. However, more than half of cells *in vivo* fire above this rate. In contrast, while CKAMP44^{-/-} neurons *in vitro* can potentially fire at close to 50Hz, they show only a very minor increase in firing frequency *in vivo* compared to the wild-type. If my interpretation of the data is correct, it would seem that some kind of compensatory mechanism is at work in the awake animal which almost completely masks the effect caused by the loss of CKAMP44.

Further, the *in vivo* experiments (ultimately on only 6 animals each) contain such an enormous number of cells that one has to question whether the statistical approach chosen can be considered valid. Would some kind of nested analysis be more appropriate here? There is clear potential for a false positive observation with samples of this size. What would the outcome have been for the control data in Tables S16 and 17 if they had N = 500+? Taken together this paper presents further evidence that the functional effects of AMPAR auxiliary proteins are of great importance to neuronal activity. However, while this work would be of potential interest, a clear and convincing explanation of the apparent differences between *in vitro* and *in vivo* recordings is clearly lacking and needs to be addressed.

Minor comments.

In general, displaying the median with IQR gives a reduced representation of the inherent variability of the recordings. This would be more transparent with box and whisker plots. p10, ln 20. Break should be brake?

Last sentence of the Discussion. Is there a typo? It doesn't seem to read properly.

Figure 4a, b. The red arrow between the cartoon and the raw data is rather confusing. It seems to indicate this is the time of the change from light to dark/dark to light. This was not helped by the different alignments of the panels below. Is the arrow necessary?

Figure 4b, raw data. 100ms axis tick is misplaced

Figure 4c, missing legend: wt is blue, KO is red

Check labelling of Supplementary tables, they seem to refer to an earlier format of the paper with 5 figures.

response to referees

We thank the Reviewers for their constructive inputs and specific suggestions, which were instrumental in designing new experiments to improve the manuscript. We performed a number of additional experiments to address all concerns that were raised by the Reviewers. One of the major criticisms raised by Reviewers 1 and 3 was the difference in the influence of CKAMP44 on firing rates of dLGN neurons *in vitro* and *in vivo*. In the first version of the manuscript, we had used a comparatively strong stimulus to activate dLGN neurons *in vitro*. We now tested different *in vitro* stimuli and found that the influence of CKAMP44 varies with frequency and number of stimuli (new Figure 4). Importantly, the influence of CKAMP44 on dLGN neuron firing rates *in vitro* was similar to that *in vivo* when using a stimulus with a frequency and stimulus number similar to that of retinal ganglion cell activities in response to full field ON- and OFF stimuli. A second major concern raised was an apparent discrepancy of the short-term plasticity of EPSCs and EPSPs. As suggested by Reviewer 1, we investigated short-term plasticity using a K^+ -containing intracellular solution. The new experiments indeed show that short-term plasticity of EPSCs and EPSPs is very similar under these conditions (Supplementary Figure 2 and 4).

We here provide a list of figure panels that were added during the revision process:

Figure 4: Firing rates of relay neurons in response to different stimulation patterns

Supplementary Figure 2: PPR of retinogeniculate synapses with K^+ -containing intracellular solution

Supplementary Figure 4: Normalized EPSCs in the presence of K^+ -containing intracellular solution

Supplementary Figure 8: Peak firing rate and rate increases of dLGN neurons plotted with $\text{mean} \pm \text{SEM}$.

Point-by-point response:

Reviewer #1 :

The authors assess the role of CKAMP44, an AMPAR auxiliary subunit, in excitatory synaptic processing by cells of the dorsal lateral geniculate (dLGN), focusing mainly on the retinogeniculate pathway. First, they show that CKAMP44 is expressed in the dLGN. They then go on to examine the functional roles of the protein in the dLGN by comparing the neurophysiology of dLGN relay neurons from control mice and from mice with genetic deletion of CKAMP44 (CKAMP44^{-/-}). There were essentially no effects on intrinsic membrane properties but there were a number of effects on AMPAR-mediated responses. AMPA responses were generally smaller in the CKAMP44^{-/-} cells, consistent with a role in AMPA receptor trafficking. There were also effects on AMPAR “gating” in retinogeniculate synapses, supporting roles for CKAMP44 in AMPAR desensitization and short-term synaptic depression. Consistent with these roles in synaptic dynamics, the authors provide evidence that deletion of CKAMP44 enhances synaptic integration and evoked firing of dLGN cells, including visually-evoked firing in awake mice.

This study is thorough and the experiments appear to be nicely done. The manuscript extensively describes the role of CKAMP44 on AMPA responses in dLGN neurons at multiple

levels of analysis. Yet, some of the experiments appear to at least partially contradict one another. Furthermore some of the analysis and writing need to be corrected so the main ideas in the manuscript are clear to the reader.

Major issues:

1. The voltage clamp experiments in Figure 2 show that the retinogeniculate synapse is depressing, as expected from decades of previous studies. CKAMP44 deletion was found to significantly weaken the short term depression, although some clear depression remained. Surprisingly, the current-clamp data in Figure 3A shows much weaker depression than would be predicted by the voltage-clamp recordings. In fact, the EPSPs from the CKAMP44 KO mice actually facilitated. The differences between the voltage-clamp and current-clamp results need to be addressed and reconciled. A major methodological difference between the voltage-clamp and current-clamp recordings involved the internal recording solutions, with cesium being used for voltage-clamp and potassium for current-clamp. It is possible that cesium, ejected onto the retinogeniculate terminals during the patching procedure, artificially enhanced short-term depression of the retinogeniculate synapses. I have personally observed such an effect of cesium in another pathway, and it required nearly an hour of rinsing for the effects of the cesium solution to dissipate following patching (and for short-term dynamics to normalize). I suggest that the authors test the excitatory synaptic currents (in voltage clamp) using a potassium-based internal solution to determine whether the differences in short-term dynamics observed in voltage-clamp and current-clamp have an underlying “neurobiological explanation” (e.g., activation of intrinsic conductances in current-clamp, leading to apparent “boosting” of synaptic responses during trains), or are instead due to a technical artefact.

We thank Reviewer 1 to make us aware of the effects of Cs on presynaptic release probability. As suggested, we recorded EPSC PPRs with a K⁺-containing intracellular solution (Supplementary Figure 2). Indeed, PPRs were increased with K⁺-containing solution in retinogeniculate synapses of both genotypes. PPRs of relay neurons in CKAMP44^{-/-} mice still significantly higher than in wildtype mice and PPRs of EPSCs are now more consistent with the observed changes of EPSP amplitudes (which is obvious when plotting relative EPSP amplitudes instead of absolute EPSP amplitudes, i.e. summation, see below). To more directly show that there is no substantial difference in the short-term plasticity of EPSCs and EPSPs, we performed an additional experiment in which we again patched relay neurons with a K⁺-containing intracellular solution and quantified the amplitudes of EPSC. In this experiment, we evoked with 10 stimulations of the optic tract at 50 Hz (thus the same protocol that we used during current clamp recording). The change in EPSC amplitudes (Supplementary Figure 4) was comparable to the changes in EPSPs amplitudes (Fig. 3). The small differences in EPSC and EPSP amplitudes are presumably explained by the activation of voltage-gated channels.

2. There is conflation of facilitation and summation. The authors often seem to be making points about summation of EPSPs but they use the term “facilitation”. Facilitation and summation are two separate phenomena. Both of these dynamic aspects of the responses could be important for a clear understanding of the roles of CKAMP44. The Methods and Results should clarify which of the phenomena are being measured and exactly how it is done.

a. In the CKAMP44 KO trace shown in Figure 3a, there are two distinct phenomena occurring:

facilitation and summation. The data analysis should account for these separately, and should be clearly explained. In order to address facilitation/depression of EPSP size, the authors should measure the peak amplitude of each EPSP relative to the baseline immediately preceding that particular EPSP. In contrast, measurements can be referenced to the baseline preceding the first stimulus in a train to assay summation of the EPSPs. The latter is what appears to be plotted in Figure 3A.

To avoid confusion of summation and facilitation, we quantified relative EPSP amplitudes relative to the baseline immediately preceding that particular EPSP, as suggested. We observed only a mild EPSP facilitation in CKAMP44^{-/-} mice and an EPSP depression in wildtype mice, consistent with the EPSC data (Fig. 3a, Supplementary Figure 5). We decided not to show the quantification of EPSP summation that we had shown in the previous version of the manuscript since we believe that the example EPSPs of Fig3a display well the EPSP amplitude summation that can be expected from the observed relative EPSP amplitudes (as also described in the results section). However, if the Reviewer thinks that showing the quantification of the summed EPSP amplitudes facilitates the understanding of the role of CKAMP44, we are of course happy to add this analysis.

b. p.7: "Current clamp experiments thus suggest that deletion of CKAMP44 increases EPSP amplitudes and firing probability of dLGN relay neurons despite the fact that it reduces synaptic AMPAR number." I believe the word "amplitudes" should be replaced with "summation" here. The initial amplitudes are decreased rather than increased, and it is unclear to me whether or not the later amplitudes are actually greater than in the WT, although the summated response seems to be enhanced.

The quantification of the relative EPSP amplitudes shows that deletion of CKAMP44 increases not only the amplitude of the second EPSP, but also EPSP 3-10 (Fig. 3a, Supplementary Figure 5). Thus, we believe that it is accurate to use the word amplitude here.

c. p.10: "This indicates that synaptic short-term facilitation of synaptic potentials in retinogeniculate synapses contributes to the processing of visual information." The normal retinogeniculate synapse has never, to my knowledge, been shown to facilitate. Instead, there can be summation at high rates.

We now show relative EPSP amplitudes that indeed display short-term depression in retinogeniculate synapses of wildtype mice, consistent with the voltage-clamp data (Fig. 3a, Supplementary Figure 5). As proposed by Reviewer 1, an increase of absolute EPSP amplitudes is observed due to summation.

3. There are a number of important mistakes in the writing, especially during summaries of the results. These mistakes could lead to real confusion. The authors should carefully revise the text with an eye to correcting such mistakes. Here are some examples:
a. p.3: "By its influence on short-term plasticity, CKAMP44 plays a role for input processing as evidenced by increased excitatory postsynaptic potential (EPSP) amplitudes and spike probability upon repetitive stimulation of the optic tract." This is confusing. I believe the authors are referring to effects of deletion of the protein, but the sentence is ambiguous and could be read to mean that the protein itself contributes to increased EPSPs upon repetitive stimulation.

On page 3 we specifically pointed out that the increased EPSP amplitudes and spike probability was observed in CKAMP44^{-/-} mice.

b. p.6: “Voltage-clamp experiments showed that CKAMP44 increases short-term depression in retinogeniculate synapses by accelerating recovery from desensitization of AMPARs.” The statement seems to be wrong; the evidence in the paper indicates that CKAMP44 slows recovery from desensitization of AMPARs rather than accelerating recovery.

This was corrected by replacing “increase” with “decrease” (see page 6).

c. p.6: “In stark contrast, current amplitudes increased steadily in synapses of CKAMP44^{-/-} mice (Fig. 3a and Table 6).” The authors are talking about synaptic potentials, not currents.

the sentence was rephrased. (see page 6)

d. p.9: “Pharmacological block of AMPAR desensitization with CTZ increases synaptic short-term depression in retinogeniculate synapses to a similar extent as the genetic deletion of CKAMP44...” Both CTZ and deletion of the protein decrease depression rather than increasing it.

“increase” was replaced with “decrease”. (see page 11).

e. p.10: “Short-term depression was indeed increased in GluA1^{-/-} mice 10, although to a much smaller extent than in CKAMP44^{-/-} mice.” Again, backward – depression was decreased in both models, correct?

“increased” was replaced with “decreased” see page 11).

f. Figure 2c Legend: The steady-state amplitude is reduced (not increased, correct?) in the CKAMP44^{-/-} mouse?

“increased” was replaced with “reduced” (see Fig. 2c Legend).

g. Figure 3c-e Legend: There does not seem to be any explanation of the panels on the right.

We added an explanation for these panels (see Fig. 3).

h. Figure S4: There are no letters labeling the distinct panels.

We added the letters (see Supplementary Figure 6).

i. The titles of the tables include references to figures, but the figure numbers seem to be wrong.

This was corrected.

Minor issues:

1. The AMPA responses are reduced upon deletion of CKAMP44, but there seems to be no direct evidence in this paper regarding surface trafficking. The authors imply that there is direct evidence for surface trafficking (e.g., in the Abstract) when there doesn't seem to be much. Please tone it down.

We changed the sentence in the abstract accordingly: "Here we identify CKAMP44 as a crucial auxiliary subunit of AMPARs in dLGN relay neurons, where it increases AMPAR-mediated current amplitudes and modulates gating of AMPARs" (see page 1).

2. In Figure 1g, there is no difference between WT and CKAMP44^{-/-} cells in terms of rates of desensitization. Instead, the differences are related to recovery from desensitization (Fig 1h) and steady-state currents. Please elaborate on the interpretation and importance of these subtleties with respect to mechanisms.

We added a paragraph in the discussion section on the mechanism that may influence recovery from desensitization and on steady-state currents. In addition, we also discussed the possible mechanism of the absence of an effect on rates of desensitization (see page 11).

3. Regarding summation seen in Figure 3A, is it possible that mGluR receptors are being activated?

There are indeed many factors that might possibly explain EPSP summation in relay neurons of wildtype and CKAMP44^{-/-} mice including EPSP kinetics, activation of voltage gated channels or NMDARs and perhaps also mGluRs. However, as discussed above, relative EPSP amplitudes are not very different from EPSC amplitudes in both genotypes if both are analyzed with a K⁺-based intracellular solution (Supplementary Figure 4). This suggests that the difference in relative and also absolute EPSP amplitudes between genotypes results mainly from the differences in AMPAR conductance and not from activation of voltage gated channels, NMDARs or mGluRs (a contribution of NMDARs was excluded by performing control experiments in the presence of APV, Supplementary Figure 5). Thus, there is no indication that CKAMP44 deletion alters other factors that result in EPSP summation such as activation of mGluRs. As the mechanisms explaining summation are not the main focus of the study, we did not discuss them in detail in the new manuscript.

4. The Introduction and/or Discussion would benefit from elaborating on the potential role of CKAMP44 on other depressing synapses in parallel systems (e.g., medial lemniscus to ventral posterior synapse and inferior colliculus to medial geniculate synapse). Do these have CKAMP44, and can the results observed in retinogeniculate synapses be generalized to any other synapses?

These are indeed very interesting questions. Our *in situ* hybridization analyses and the Allen Brain Atlas show a strong CKAMP44 mRNA signal in several thalamic nuclei including the medial geniculate nucleus and the ventral posterior nucleus. Thus, it seems likely that CKAMP44 plays a role in pronounced short-term depression (Arsenault, D. and Zhang, Z.W, *The Journal of Physiology*, 2006; Bartlett, E.L. and Smith, P.H, *Neuroscience*, 2002) also in these thalamic nuclei. Another interesting example is the mediodorsal thalamus where CKAMP44 mRNA appears to be expressed and where the excitatory input from the paleocortical piriform

cortex displays strong paired-pulse depression with a high release probability. In addition, axons from the paleocortical piriform cortex form giant synapses with multiple synaptic contacts and strong single fiber input (approx. 400 pA) onto neurons from the mediodorsal thalamus (Groh et al., *The Journal of Neuroscience*, 2008). Thus, there are some resemblances to retinogeniculate synapses. It is thus possible that CKAMP44 generally modulates the excitatory input in thalamic neurons that receive sensory input and may serve to prevent their overactivation. In fact, we aim at investigating the role of CKAMP44 in other thalamic neurons in future studies.

We discussed these considerations in the revised manuscript (see discussion, page 12).

5. For figure 2 and 3: Comment on use of naturally occurring firing patterns and frequencies. What stimulus frequencies correspond to typical RGC rates? Please elaborate on both “background” firing rates and on stimulus-evoked rates.

Our aim was to use *in vitro* stimulus frequencies that are in the range of naturally occurring firing frequencies. The 50 Hz stimulus would correspond to RGC activity in response to strong visual stimuli. In the revised manuscript, we added additional experiments, in which we investigated also the influence of CKAMP44 to stimuli with lower frequencies. *In vivo* recordings of mouse ON and OFF RGCs showed peak firing rates of 20 Hz in response to the presentation of full-field ON- or OFF-stimuli. RGCs fire with “background” rates of approximately 1-10 Hz when the intensity of the monitor is at mean average luminance (Sagdullaev, B.T. and McCall, M.A., *Vis Neurosci* 2005). Thus, RGC firing frequencies are in the range of frequencies with which we stimulated the optic tract in the new experiments presented in Figure 4. A 25 Hz stimulus, which is similar to the peak firing frequency of RGCs during the presentation of full-field ON- or OFF-stimuli, elicited *in vitro* a response of dLGN neurons with a relatively small difference in firing rates between genotypes. This difference is indeed comparable to the difference in peak firing rates of dLGN neurons *in vivo* when mice are displayed with full-field ON- or OFF-stimuli. Presentation of high or low luminance spots of optimal size can elicit responses with peak firing rates larger than 50 Hz, thus considerably higher than full-field ON- or OFF-stimuli (Sagdullaev, B.T. and McCall, M.A., *Vis Neurosci* 2005). RGC peak firing frequencies can even transiently reach up to 500 Hz during presentation of natural visual stimuli (Zeck et al. *Eur J Neurosci*, 2007). The *in vitro* analyses showed that the influence of CKAMP44 on relay neuron firing rates increases with stimulation frequency, suggesting that CKAMP44 plays a bigger role for input integration when mice are presented with visual stimuli that elicit higher RGC firing rates than full-field ON or OFF-stimuli. Absolute firing rates and differences of firing rates between genotypes were also dependent on stimulation strength and number of stimuli (see Figure 4). We did not find any publication in which the number of action potentials in response to ON- and OFF-stimuli were investigated in mice with *in vivo* recordings. However, RGCs fire approximately 5 spikes/per coding event in response to the presentation of visual stimuli such as moving gratings. Very strong visual stimuli such as flashed spots elicited even 10 spikes/per coding event (Zeck et al. *Eur J Neurosci*, 2007). Of note, there was a small, but significant differences in firing rates between genotypes when stimulating the optic tract *in vitro* with 5 stimuli and a strongly significant difference with 10 stimuli irrespective of the stimulation frequency (Figure 4a-c).

We added these considerations to the discussion of the revised manuscript (see discussion, page 14).

6. The Methods indicate that the *in vitro* recordings were performed at room temperature.

Was this true even for the functional measures of synaptic integration and spiking? Were results different when recordings were done closer to physiological temperatures? This seems to be important considering the rather weak effects observed in vivo (Fig. 4). Please discuss.

Indeed, release probability, glutamate reuptake, spillover glutamate and AMPAR gating properties are all temperature-dependent. Release probability and glutamate transporter activity increase with temperature (Asztely, F., Erdemli, G. and Kullmann, D.M, *Neuron*, 1997; Volgushev, M., *et al.*, *Journal of Neurophysiology*, 2004; Postlethwaite, *et al.*, *The Journal of Physiology*, 2007). The faster glutamate reuptake decreases the level of glutamate spillover at physiological temperature when compared to room temperature (Asztely, F., Erdemli, G. and Kullmann, D.M, *Neuron*, 1997). High release probability should increase the influence of CKAMP44 on short-term plasticity, whereas low levels of spillover glutamate should have the opposite effect. In fact, CKAMP44 influences short-term plasticity when recordings are performed at room temperature (in retinogeniculate synapses; this study) and at physiological temperature (in perforant path-to-granule cell synapses; Khodosevich, K., *et al. Neuron*, 2014). In the dentate gyrus, we analyzed the influence of CKAMP44 on short-term plasticity also at room temperature, and found very similar PPRs when compared to the PPRs obtained at physiological temperature. Recording from dLGN neurons at physiological temperature was more difficult than at room temperature, which is why we performed the *in vitro* experiments of this study at room temperature. Differences in temperature may partly explain the different influence of CKAMP44 *in vitro* and *in vivo*. However, we believe that differences in relay neuron firing rates are more likely explained by differences in RGC firing rates *in vivo* and optic tract stimulation frequencies *in vitro* (see previous response and page 12).

7. Methods and/or Results section: elaborate on stimulus intensity. Are there any differences in the stimulus intensities used per genotype?

There was no obvious difference in the stimulus intensity, in particular not in current clamp experiments in which we adjusted stimulus strength such that the 1st EPSP amplitude was smaller in relay neurons of CKAMP44^{-/-} than of wildtype mice (reflecting the difference in AMPA/NMDA ratio). However, the stimulus intensities that result in a similar EPSC or EPSP amplitude varied substantially by a factor of 100 also within a genotype (most likely depending on how much of the optic tract was severed), such that we did not try to compare stimulus intensities between genotypes.

8. Abstract: Change “know” to “known”

“known” was changed to “known” (see Abstract page 1).

9. Figure 1e, Contol should be Control

“Contol” was changed to “Control” (see Fig.1e).

10. Results section for figure 4: Electroretinogram data in Figure s6 not s7.

This was corrected.

11. Figure s2: 43% not 44%. Discrepancy between figure and text.
"44%" was changed to "43%" (see Fig.3).

12. As far as I can tell, reference 36 does not actually describe the nucleated patch technique although it does seem to use the method. Please provide a reference that describes the technique.

We now referenced the correct paper: Sather et al., The Journal of Physiology 1992.

Reviewer #2:

This electrophysiological study provides new insights regarding the regulation of AMPA receptors by the CKAMP44 auxiliary subunit. The authors specifically focus on retinogeniculate synapses because these are characterized by a robust short-term depression of AMPA receptor mediated currents and the mechanism is unknown. Through a series of rigorous studies the author demonstrate that CKAMP44 mediates this short term plasticity via its reducing the rate of AMPA receptor recovery from desensitization. Because the physiological roles for CKAMP44 and for AMPA receptor desensitization are poorly understood, this paper is of general interest.

The authors first show that CKAMP44 is expressed in dLGN neurons, where it controls the gating and amplitude of AMPA receptor mediated currents. These data are clear and convincing.

The authors next show that that CKAMP44 specifically modulates the paired pulse ratio in reticulogeniculate synapses but not in corticogeniculate synapses. Again, these experiments are well done and the results are clear.

The authors show that CKAMP44 decreases the firing probability of dLGN neurons. This is evidenced by many experiments such as the increase in EPSP amplitudes and the increase in spike probability during a 10x50 Hz stimulus in CKAMP44 KO mice.

Finally, the authors argue that CKAMP decreases peak firing rate in ON and OFF responses in vivo. These data are less convincing. Especially difficult to interpret are data in figure 4c, which quantifies the ON and OFF responses of dLGN neurons. The data are presented as median +/- IQR. The small differences between WT and KO and the very large IQRs make it hard to appreciate that the changes are significant. It would be important to see the individual data points and to have other statistics such as the S.E.M.

We replotted the data in Supplementary Figure 8 as mean \pm SEM (we also still show the data in a box whisker plot with median and IQR as requested by Reviewer 3). The presentation of data in Supplementary Figure 8 shows that CKAMP44 deletion robustly and significantly increases mean peak rates by 10% and 28% and rate increases by 39% and 31% for ON and OFF cells, respectively. This is indeed less obvious when looking at the box-whisker plotted data with the very large IQRs. We think that showing individual data points is not too helpful for facilitating the interpretation of data due to the large number of neurons (ON-responding cells: 542 wildtype and 430 CKAMP44^{-/-} neurons, OFF-responding cells: 309 wildtype and 285

CKAMP44^{-/-} neurons) and the skewed data distribution (see figure below). However, if the Reviewer thinks differently, we will be happy to add also the individual data points.

The difference in ON and OFF rates between genotypes is smaller than that observed during *in vitro* recording when using strong stimulus protocols (i.e. high stimulus number and frequency). We added new experiments that show that the influence of CKAMP44 varies with frequency and number of action potentials of retinal ganglion cells (Fig. 4). When using *in vitro* stimulation protocols with frequencies of 25 Hz, which is comparable to that of RGCs during presentation of full-field ON- and OFF-stimuli (Sagdullaev, B.T. and McCall, M.A. *Visual Neuroscience*, 2005), we observed differences in firing rates between genotypes that were in the range of what we observed *in vivo*. (see Supplementary Figure 8 and discussion page 13 and 14).

Reviewer #3:

This manuscript reports a functional role for the AMPAR auxiliary protein CKAMP44 in the modulation of AMPAR currents in the lateral geniculate nucleus (LGN). While TARPs have received the majority of attention in the relatively new field of AMPAR auxiliary subunits, we still know comparatively little about CKAMP44. It has been established in previous work by the senior author that the slowing of AMPAR recovery from desensitization by CKAMP44 can influence STP in the dentate gyrus. The current work reports a similar finding in the LGN, and goes further by presenting in vivo data suggesting that in a CKAMP44^{-/-} mouse line, visual processing is modified. This is the first report of an in vivo role for an auxiliary subunit that slows recovery from desensitization. As such it represents a good advance in our understanding. The data are of high quality, as we have come to expect from this lab, and the paper is well structured. However, there are issues which are a concern, and which clearly need to be addressed.

My major concern is that the in vivo changes, which are critical to the novelty of the study, appear to be of a very different magnitude to those detected in slice recordings. The data in Figures 3 and 4 seem poorly comparable and this does not appear to have been given due consideration in the study or the presentation. The obvious concern is whether the firing patterns elicited in Figure 3 are of relevance to the situation in vivo. Specifically, Figure 3

seems to demonstrate that wild-type cells should rarely be able to fire above 10 Hz despite direct stimulation. However, more than half of cells in vivo fire above this rate. In contrast, while CKAMP44^{-/-} neurons in vitro can potentially fire at close to 50Hz, they show only a very minor increase in firing frequency in vivo compared to the wild-type. If my interpretation of the data is correct, it would seem that some kind of compensatory mechanism is at work in the awake animal which almost completely masks the effect caused by the loss of CKAMP44.

This concern is indeed very relevant. As suggested by the Reviewer, the difference in the influence of CKAMP44 on relay neuron firing frequencies *in vitro* and *in vivo* can be explained by the *in vitro* stimulation pattern that is different to the firing patterns of retinal ganglion cells *in vivo*. We performed several new *in vitro* experiments to address the question of the dependence of the influence of CKAMP44 on the stimulus pattern (Fig. 4). The new data show that the influence of CKAMP44 on relay neuron firing rates increases with frequency and number of optical tract stimulations *in vitro*. In the response to Reviewer 1 (minor concern 5) we addressed the question of how comparable the *in vitro* stimuli are to the retinal ganglion cell firing frequencies *in vivo* (see above). Importantly, firing rates observed during *in vitro* recordings depend not only on frequency and number of optical tract stimulations, but also on the stimulation strength. For the investigation of the influence of CKAMP44 on firing rates *in vitro* (Fig. 3b-e and new experiments shown in Fig. 4a-c), we used a weak stimulation strength that did not elicit an action potential during the first stimulus. The reason for this was to be able to adjust the stimulation such that the first EPSP was approximately 43% smaller in neurons of CKAMP44^{-/-} mice than in those of wildtype mice with the idea that we activate in this case a similar number of axons (AMPA/NMDA ratios were reduced by 43% in CKAMP44^{-/-} mice). For the investigation of the dependence of the influence of CKAMP44 on the stimulus frequency and stimulus number (Fig. 4) we used a stimulation strength that evoked similar firing rates in wildtype neurons (Fig. 4c, 50 Hz stimulus paradigm) as compared to the *in vivo* peak firing rates (Fig. 5). With this stimulation paradigm, firing rates were different between genotypes when 5 or more stimuli were applied. We did not find any publication in which the number of action potentials in response to ON- and OFF-stimuli were investigated in mice with *in vivo* recordings. However, RGCs fire approximately 5 spikes/per coding event in response to the presentation of visual stimuli such as moving gratings. Very strong visual stimuli such as flashed spots even elicited 10 spikes/per coding event (Zeck et al. *Eur J Neurosci*, 2007). Of note, there was a small, but significant difference in firing rates between genotypes when stimulating the optic tract *in vitro* with 5 stimuli and a bigger difference with 10 stimuli irrespective of the stimulation frequency (Figure 4a-c). We have now added yet another experiment, in which the stimulation strength was suprathreshold for the first stimulus. This would reflect a very strong visual stimulus (e.g. in response to a spot illumination) that recruits several retinal ganglion cells, which fire at high frequency with many action potentials (see above). Firing frequencies of neurons in both genotypes were consequently much higher than the frequencies of neurons observed *in vivo* with ON/OFF visual stimulation. However, there was also a significant difference in firing rates between genotypes (Fig. 4d).

Further, the in vivo experiments (ultimately on only 6 animals each) contain such an enormous number of cells that one has to question whether the statistical approach chosen can be considered valid. Would some kind of nested analysis be more appropriate here? There is clear potential for a false positive observation with samples of this size. What would the outcome have been for the control data in Tables S16 and 17 if they had N = 500+?

Taken together this paper presents further evidence that the functional effects of AMPAR auxiliary proteins are of great importance to neuronal activity. However, while this work would be of potential interest, a clear and convincing explanation of the apparent differences between in vitro and in vivo recordings is clearly lacking and needs to be addressed.

The only statistically significant differences in the *in vivo* experiment are when comparing the peak rate response and percentage of rate increase to visual stimulation. The number of cells in each group was as follows.

ON-responding cells: 542 wildtype and 430 CKAMP44^{-/-} neurons

OFF-responding cells: 309 wildtype and 285 CKAMP44^{-/-} neurons

A Wilcoxon rank sum test was used for these comparisons because it does not assume that the data are normally distributed and that the groups have equal variance. In addition, this test is less likely to generate significant results because of the outliers.

One strategy could have been to use a mixed model nested ANOVA with the fixed factor genotype and the random nested factor mouse. This would require the data to be normally distributed and the variance to be equal in the two genotypes. Both of these assumptions were violated by the data (Shapiro-Wilk normality test: $P < 10^{-16}$ for both dependent variables; Bartlett test of homogeneity of variances: $P < 10^{-16}$ for both dependent variables). For these reasons, we decided against using an nested ANOVA.

The reviewer correctly pointed out that results of some of the control experiments could be significantly different between genotypes with a higher sample size. Data in Table S16 and 17 (in the new manuscript Supplementary Table 22 and 23) showed that CKAMP44 deletion leads to a non-significant reduction in mEPSC amplitude (7%), in a-wave amplitude (17%) and b-wave amplitude (8%). We performed a power analysis to address the question if the data in Tables S16 and 17 would be significantly different between genotypes if the n was >500 . For this analysis we assumed means and distributions that are the same as in Supplementary Table 22 and 23. Indeed, an N of >500 would suffice to detect statistical differences that are in the range of the observed differences in mEPSC amplitude (7% smaller in CKAMP44^{-/-} mice), the a-wave amplitude (17% smaller in CKAMP44^{-/-} mice) and the b-wave amplitude (8% smaller in CKAMP44^{-/-} mice). There would be no statistical difference in pERG amplitudes at preferred spatial frequencies. Thus, it is possible that we would detect statistical differences in the control experiments with a high n . However, smaller mEPSC, a-wave and b-wave amplitudes in CKAMP44^{-/-} mice would not explain the increased firing frequency of dLGN neurons in response to visual stimulation. On the contrary, one would expect smaller firing rates and rate increases of dLGN neurons in CKAMP44^{-/-} mice than in wildtype mice. Finally, the differences of firing rates (median: 10% and 34% higher ON and OFF- responses, respectively, in CKAMP44^{-/-} mice, mean: 10% and 28%) and rate increases (median: 16% and 49% higher ON- and OFF- responses, respectively, in CKAMP44^{-/-} mice, mean: 39% and 31%) are actually not that small and comparable to the differences in firing rates of relay neurons *in vitro* after the 5th stimulus when stimulating with 50 Hz. Bigger differences in firing rates are indeed observed with more stimuli and higher stimulation strength (Fig. 4), which would correspond to the activation of relay neurons in response to presentation of optimal stimuli (e.g. spot stimuli).

In general, displaying the median with IQR gives a reduced representation of the inherent variability of the recordings. This would be more transparent with box and whisker plots.

We now show the data as box and whisker plots for all non-normal distributed data (Figure 1, Figure 5, Supplementary Figure 1, Supplementary Figure 5, Supplementary Figure 6, Supplementary Figure 7, Supplementary Figure 9).

p10, In 20. Break should be brake?

This was corrected (see discussion, page 12).

Last sentence of the Discussion. Is there a typo? It doesn't seem to read properly. Figure 4a, b. The red arrow between the cartoon and the raw data is rather confusing. It seems to indicate this is the time of the change from light to dark/dark to light. This was not helped by the different alignments of the panels below. Is the arrow necessary?

We rephrased this sentence (see discussion, page 14).

Figure 4b, raw data. 100ms axis tick is misplaced

100 ms axis was replaced (Fig 5b).

Figure 4c, missing legend: wt is blue, KO is red

We added inset into that figure (Fig, 5).

Check labelling of Supplementary tables, they seem to refer to an earlier format of the paper with 5 figures.

This was corrected.

REVIEWERS' COMMENTS:

Reviewer #1 (Remarks to the Author):

The authors have addressed all of the issues and criticisms raised in my previous review. They conducted substantial new experiments and analysis, and revised the text adequately. It is an interesting paper with well-justified claims. Nice work.

Reviewer #2 (Remarks to the Author):

The authors have appropriately address my concerns and I believe the paper is now appropriate for publication.

Reviewer #3 (Remarks to the Author):

I think the authors have made a good attempt to address my main concerns. This is now an interesting, thorough and valuable contribution to the field.

Response to referees

we thank the reviewers for their intensive reading and careful review. Since they did not raise new criticisms or suggestions, we did not include any major changes for this revision.

Point-by-point response:

Reviewer #1 :

The authors have addressed all of the issues and criticisms raised in my previous review. They conducted substantial new experiments and analysis, and revised the text adequately. It is an interesting paper with well-justified claims. Nice work.

No additional issue.

Reviewer #2

The authors have appropriately address my concerns and I believe the paper is now appropriate for publication.

No additional issue.

Reviewer #3

I think the authors have made a good attempt to address my main concerns. This is now an interesting, thorough and valuable contribution to the field.

No additional issue.